# A Decade of GOSAT Proxy Satellite CH$_4$ Observations

Robert J. Parker[1,2], Alex Webb[1,2], Hartmut Boesch[1,2], Peter Somkuti[3], Rocio Barrio Guillo[1,2], Antonio Di Noia[2], Nikoleta Kalaitzi[1,2], Jasdeep S. Anand[2], Peter Bergamaschi[4], Frederic Chevallier[5], Paul I. Palmer[6,7], Liang Feng[6,7], Nicholas M. Deutscher[8], Dietrich G. Feist[9,10,11], David W. T. Griffith[8], Frank Hase[12], Rigel Kivi[13], Isamu Morino[14], Justus Notholt[15], Young-Suk Oh[16], Hirofumi Ohyama[14], Christof Petri[15], David F. Pollard[17], Coleen Roehl[18], Mahesh K. Sha[19], Kei Shiomi[20], Kimberly Strong[21], Ralf Sussmann[22], Yao Té[23], Voltaire A. Velazco[8], Thorsten Warneke[15], Paul O. Wennberg[18], and Debra Wunch[24]

[1]National Centre for Earth Observation, University of Leicester, UK

[2]Earth Observation Science, School of Physics and Astronomy, University of Leicester, UK

[3]Cooperative Institute for Research in the Atmosphere, Colorado State University, Colorado, US

[4]European Commission Joint Research Centre, Ispra (Va), Italy

[5]Laboratoire des Sciences du Climat et de L'Environnement, LSCE/IPSL, CEA-CNRS-UVSQ, Université Paris-Saclay, Gif Sur Yvette, France

[6]School of GeoSciences, University of Edinburgh, Edinburgh, Scotland

[7]National Centre for Earth Observation, University of Edinburgh, Edinburgh, Scotland

[8]Centre for Atmospheric Chemistry, School of Earth, Atmospheric and Life Sciences, University of Wollongong

[9]Ludwig-Maximilians-Universität München, Lehrstuhl für Physik der Atmosphäre, Munich, Germany

[10]Deutsches Zentrum für Luft- und Raumfahrt, Institut für Physik der Atmosphäre, Oberpfaffenhofen, Germany

[11]Max Planck Institute for Biogeochemistry, Jena, Germany

[12]Karlsruhe Institute of Technology, IMK-ASF, Karlsruhe, Germany

[13]Space and Earth Observation Centre, Finnish Meteorological Institute, Finland

[14]National Institute for Environmental Studies (NIES), Tsukuba, Japan

[15]Institute of Environmental Physics, University of Bremen, Germany

[16]Climate Research Division, National Institute of Meteorological Sciences (NIMS), Jeju-do 63568, Republic of Korea

[17]National Institute of Water and Atmospheric Research Ltd (NIWA), Lauder, New Zealand

[18]California Institute of Technology, Pasadena, CA 91125, USA

[19]Royal Belgian Institute for Space Aeronomy (BIRA-IASB), Brussels, Belgium

[20]Earth Observation Research Center, Japan Aerospace Exploration Agency, Japan

[21]Department of Physics, University of Toronto, Canada

[22]Karlsruhe Institute of Technology, IMK-IFU, Garmisch-Partenkirchen, Germany

[23]Laboratoire d'Etudes du Rayonnement et de la Matière en Astrophysique et Atmosphères (LERMA-IPSL), Sorbonne Université, CNRS, Observatoire de Paris, PSL Université, 75005 Paris, France

[24]Department of Physics, University of Toronto, Toronto, Canada

**Correspondence:** R. J. Parker (rjp23@le.ac.uk)

**Abstract.** This work presents the latest release (v9.0) of the University of Leicester GOSAT Proxy $XCH_4$ dataset. Since the launch of the GOSAT satellite in 2009, this data has been produced by the UK National Centre for Earth Observation (NCEO) as part of the ESA Greenhouse Gas Climate Change Initiative (GHG-CCI) and Copernicus Climate Change Services (C3S) projects. With now over a decade of observations, we outline the many scientific studies achieved using past versions of this data in order to highlight how this latest version may be used in the future.

We describe in detail how the data is generated, providing information and statistics for the entire processing chain from the L1B spectral data through to the final quality-filtered column-averaged dry-air mole fraction ($XCH_4$) data. We show that out of the 19.5 million observations made between April 2009 and December 2019, we determine that 7.3 million of these are sufficiently cloud-free (37.6%) to process further and ultimately obtain 4.6 million (23.5%) high-quality $XCH_4$ observations. We separate these totals by observation mode (land and ocean sun-glint) and by month, to provide data users with the expected data coverage, including highlighting periods with reduced observations due to instrumental issues.

We perform extensive validation of the data against the Total Carbon Column Observing Network (TCCON), comparing to ground-based observations at 22 locations worldwide. We find excellent agreement to TCCON, with an overall correlation coefficient of 0.92 for the 88,345 co-located measurements. The single measurement precision is found to be 13.72 ppb and an overall global bias of 9.06 ppb is determined and removed from the Proxy $XCH_4$ data. Additionally, we validate the separate components of the Proxy (namely the modelled $XCO_2$ and the $XCH_4/XCO_2$ ratio) and find these to be in excellent agreement with TCCON.

In order to show the utility of the data for future studies, we compare against simulated $XCH_4$ from the TM5 model. We find a high degree of consistency between the model and observations throughout both space and time. When focusing on specific regions, we find average differences ranging from just 3.9 ppb to 15.4 ppb. We find the phase and magnitude of the seasonal cycle to be in excellent agreement, with an average correlation coefficient of 0.93 and a mean seasonal cycle amplitude difference across all regions of -0.84 ppb.

This data is available at https://doi.org/10.5285/18ef8247f52a4cb6a14013f8235cc1eb (Parker and Boesch, 2020).

# 1 Introduction

Atmospheric methane ($CH_4$) is the second most important greenhouse gas in terms of anthropogenic climate radiative forcing (Myhre et al., 2013) with a global warming potential on a 100-year time-scale of 28-34 times that of $CO_2$ (Etminan et al., 2016) on a mass/mass basis. This strong warming potential, when coupled to its short lifetime relative to that of $CO_2$ (Prather et al., 2012) makes it of particular interest when considering rapid and achievable mitigation strategies (Nisbet et al., 2020).

Scientific debate continues on trying to explain the atmospheric $CH_4$ trend observed over the past couple of decades. Records from surface sites reveal a plateau from 2000 to 2007 and a resumed increase after 2007 (Rigby et al., 2008; Dlugokencky et al., 2009). Amongst the varied surface sources of $CH_4$, the largest are natural wetlands, agriculture, livestock, biomass burning, waste and fossil fuel production; whereas the primary sink is the OH radical in the atmosphere (Kirschke et al., 2013; Saunois et al., 2020). Various hypotheses have been offered that attempt to attribute the behaviour in the global growth rate to a particular component or mechanism (Nisbet et al., 2016; Schaefer et al., 2016; Hausmann et al., 2016; McNorton et al., 2016b; Buchwitz et al., 2017a; Worden et al., 2017; Turner et al., 2017; Rigby et al., 2017) but currently there is no consensus within the community. Many of these studies have utilised satellite observations of atmospheric $CH_4$ and have shown the increasing capability of such measurements to characterise global and regional surface methane fluxes (Jacob et al., 2016).

This work presents the most recent update to the University of Leicester GOSAT Proxy $XCH_4$ Retrieval. This version (v9.0) now provides over a decade of global total column $CH_4$ observations, from April 2009 to December 2019. A full reprocessing of the entire time series has been performed to ensure consistency throughout the record and to ensure that results utilising the entire record are as robust as possible.

It is the intention of the authors that this study acts as a reference for everyone making use of the data and as such, we have attempted to provide as much practical detail as possible on the usage of the data.

Section 2 describes the GOSAT observations themselves and highlights any instrument anomalies or data gaps. Section 3 is broken down into several sub-sections detailing the usage of previous versions of this data by the scientific community. Section 4 gives an overview of the retrieval method and details the end-to-end data processing chain including statistics on throughput and data availability. Section 5 shows the validation of the data against the Total Carbon Column Observing Network, characterising not only the final Proxy $XCH_4$ data but also the individual components of the retrieval. Section 6 provides details of the global distribution of the data. Section 7 further characterises the data by performing model comparisons at global and regional scales. Finally we provide a summary and recommendations for future use in Section 8.

# 2 GOSAT TANSO-FTS Observations

GOSAT was launched in January 2009 by the Japanese Space Agency (JAXA), as the first satellite mission dedicated to making greenhouse gas observations (Kuze et al., 2009). GOSAT is nicknamed *"Ibuki"*, meaning *breath* in Japanese, highlighting that its mission involves monitoring the breathing of the planet, through measurement of the carbon cycle. In order to achieve this, GOSAT is equipped with a high-resolution Fourier Transform Spectrometer (TANSO-FTS - Thermal And Near infrared Sensor for carbon Observations – Fourier Transform Spectrometer). Shortwave infrared bands at 0.76 μm ($O_2$), 1.6 μm ($CO_2$

and $CH_4$) and 2.0 µm ($CO_2$) all provide near-surface sensitivity while a thermal infrared band between 5.5 and 14.3 µm provides mid-tropospheric sensitivity.

The objective for GOSAT was to provide routine measurements appropriate for regional and continental-scale flux estimates. Kuze et al. (2009) state a target relative accuracy of 2% for $CH_4$ over 3 month averages at 100-1000 km spatial scales. The ESA GHG-CCI User Requirements Document (URD) specify goal (G), breakthrough (B) and threshold (T) requirements, with the goal requirements being the most stringent (Buchwitz et al., 2017a). For $XCH_4$ these values are 9, 17 and 34 ppb respectively for precision and 1, 5 and 10 ppb for relative accuracy. We discuss in Section 5 how we exceed these breakthrough requirements.

As an FTS acquisition is relatively slow ($\sim$4 seconds), the GOSAT sampling strategy is tailored to achieve this goal by measuring with a relatively large footprint of 10.5 km, spaced apart approximately $\sim$263 km across-track and $\sim$283 km along-track. This means that while GOSAT does not "image" the surface, it does return to the same location every 3 days allowing a long time series of comparable measurements to be obtained. As well as nominally measuring over land in nadir mode, GOSAT is also capable of measuring over the ocean, which is normally too dark in the SWIR. This is achieved in the so-called "ocean sun-glint" observation mode, when the sun-satellite angle allows for a sufficiently reflected signal from the glint spot. Discussion of the GOSAT measurement sampling strategy and its evolution over time can be found in Appendix E.

## 2.1 Instrument Anomalies and Data Gaps

Throughout its 10 years of operation, GOSAT has experienced a number of incidents resulting in instrument anomalies (Kuze et al., 2016). These incidents include:

- May 2014 - a solar paddle incident resulting in a temporary instrument shutdown.

- January 2015 - a switch to the secondary pointing mechanism due to degradation of the primary system.

- August 2015 - a cryocooler shutdown and restart.

- May 2018 - a CDMS (Command and Data Management System) incident resulting in GOSAT being inactive for 2 weeks.

- November 2018 - rotation anomaly of the second solar paddle.

The temporary reduction in observations related to these incidents is discussed in Section 4.3 and reflected in Figure 4.

## 3 Studies Utilising Proxy $XCH_4$ Data

The University of Leicester GOSAT Proxy $XCH_4$ data is produced operationally for the ESA Greenhouse Gas Climate Change Initiative (Buchwitz et al., 2017b) and the Copernicus Climate Change Service (C3S) (Buchwitz et al., 2018) as well as routinely for the UK National Centre for Earth Observation. This work details Version 9.0 of the University of Leicester

GOSAT Proxy XCH$_4$ data but previous versions over the past decade have been used for a wide variety of scientific studies. This section details some of these past studies in order to highlight the potential applications for this data.

## 3.1 Validation of data

Firstly, before any conclusions can be drawn from analysis of the data, the data itself must be validated to ensure its robustness and reliability. Previous versions of the data have been extensively validated against the TCCON network (Total Carbon Column Observing Network) as part of the ESA Climate Change Initiative (Parker et al., 2011; Dils et al., 2014; Buchwitz et al., 2017a), including extensive validation of the model XCO$_2$ used in the generation of the data (Parker et al., 2015). We have also performed validation of the data against aircraft profile observations over the Amazon (Webb et al., 2016), one of the most important and challenging regions for the retrieval.

## 3.2 Comparison to other satellite observations

Although GOSAT was the first satellite mission dedicated to measuring GHGs, successful CH$_4$ retrievals were performed previously from SCIAMACHY and continue to be performed from new missions such as TROPOMI onboard Sentinel 5-Precursor and the recently launched GOSAT-2. Furthermore, many thermal infrared missions are capable of measuring CH$_4$ (IASI, AIRS, TES, CrIS), albeit with sensitivity to the mid-troposphere and little sensitivity to the surface. Nevertheless, it is important that these different observations are consistent and their capabilities well-understood if we wish to perform long-term analysis. The ESA Greenhouse Gas Climate Change Initiative (ESA GHG-CCI) (Buchwitz et al., 2017a) made substantial efforts to characterise and validate these different observations (Dils et al., 2014). The ensemble median algorithm (EMMA) (Reuter et al., 2020) homogenises the SCIAMACHY and GOSAT datasets produced via the ESA GHG-CCI project and is intended to be a long time series dataset for climate applications.

Studies such as Cressot et al. (2014) and Alexe et al. (2015) have investigated the consistency between flux inversions utilising SCIAMACHY, GOSAT (and IASI) CH$_4$ observations and generally found good consistency in derived emissions. Worden et al. (2015) combined the surface sensitivity of GOSAT with the mid-tropospheric sensitivity of the NASA TES instrument to better estimate the lower tropospheric methane (and hence surface) emissions while Siddans et al. (2017) have compared their IASI CH$_4$ product to GOSAT observations, finding good consistency between the two.

## 3.3 Investigation of the global growth rate

Perhaps the most important scientific question related to atmospheric CH$_4$ concentrations is understanding the observed long-term behaviour. The cause of the so-called "hiatus" or plateau in atmospheric CH$_4$ between 2000 and 2007 remains unresolved with various studies speculating on the reason. Although the GOSAT record unfortunately only began in 2009, after the end of the plateau period, it can still help to characterise behaviour and understand the processes that may have contributed to the stalling. GOSAT data have been successfully used to infer long-term global fluxes. As well as contributing to the Global Methane Budget assessments (Saunois et al., 2016, 2020), GOSAT data have been used to assess the role of regional wetland

emissions (McNorton et al., 2016b; Maasakkers et al., 2019) and the role of OH variability as a potential cause for the stalling in growth rate (McNorton et al., 2016a; Maasakkers et al., 2019).

## 3.4 Regional emissions

GOSAT data have been successfully utilised in regional-scale studies to determine $CH_4$ fluxes over many different regions. These types of studies are of particular interest as they can help inform policy-related discussions on validation and verification of regional or country-scale emission targets, such as those relevant to the Paris Agreement (Bergamaschi et al., 2018a). Fraser et al. (2013) performed regional flux inversions and found large changes over Temperate Eurasia and Tropical Asia, with the satellite observations providing a significant error reduction over only using surface data. Wecht et al. (2014) performed continental-scale inversions over North America and produced estimates of Californian $CH_4$ emissions and found consistent emission estimates over the Los Angeles Basin between the satellite inversion and that from a dedicated aircraft campaign. Turner et al. (2015) extended this work to the entire US and inferred a US anthropogenic $CH_4$ source over 50% larger than that from EDGAR and EPA bottom-up inventories. Satellite inversion results from Alexe et al. (2015) showed a redistribution of $CH_4$ emissions in the US from the north-east to south-central. These results are consistent with recent independent studies that suggest that bottom-up estimates of North American fossil fuel emissions (particularly related to natural gas and petroleum production facilities) are systematically underestimated. Ganesan et al. (2017) used GOSAT data to infer India's $CH_4$ emissions between 2010-2015 and found average emissions of 22.0 Tg yr$^{-1}$ to be consistent with the emissions reported by India to the UNFCCC with no significant trend over time. Sheng et al. (2018) performed a similar study over the US, Canada and Mexico and found that US emissions increased by 2.5% over the 7-year study period and attributed this to contributions from the oil and gas industry and livestock. In Feng et al. (2017) the individual $XCO_2$ and $XCH_4$ components from the Proxy retrieval are used to infer regional $CO_2$ and $CH_4$ fluxes simultaneously. Finally, Lunt et al. (2019) inferred $CH_4$ emissions over tropical Africa and found a linear increase of between 1.5 and 2.1 Tg yr$^{-1}$ for 2010-2016, attributing much of this to short-term increase in emissions over the Sudd wetland area in South Sudan.

## 4 UoL Proxy XCH$_4$ Retrieval

The University of Leicester Full-Physics retrieval algorithm (UoL-FP) is based on the original NASA Orbiting Carbon Observatory (OCO) "Full-Physics" retrieval algorithm (Connor et al., 2008; Boesch et al., 2011; O'Dell et al., 2012) which was designed to simultaneously fit the short-wave infrared radiances in the 0.76 μm $O_2$-A band and the 1.6 μm and 2 μm $CO_2$ bands. This algorithm has been adapted for use on GOSAT observations and modified to perform a variety of different retrievals, including the Proxy method described here. The radiative transfer calculations are accelerated using the Low-Stream Interpolation approach (O'Dell, 2010).

The concept behind the Proxy $XCH_4$ retrieval approach (Frankenberg et al., 2006) is that the majority of atmospheric scattering and instrument effects will be similar for $CH_4$ and $CO_2$ mole fraction retrievals performed in a common absorption band (around 1.6 μm, where both $CO_2$ and $CH_4$ have absorption features). By taking the ratio of the retrieved $XCH_4$/$XCO_2$,

the $CO_2$ acts as a "proxy" for the modifications to the light path induced by scattering (Butz et al., 2010) and cancels out those in the $CH_4$ retrieval. This means that moderate aerosol scattering does not adversely impact upon the retrieval, resulting in a higher number of high-quality $XCH_4$ observations compared to the full-physics approaches where much stricter post-filtering are often required. This is especially useful over the tropics, where moderate aerosol or cirrus effects can limit the coverage of full-physics methods but affect the proxy approach far less severely.

In order to convert the retrieved $XCH_4/XCO_2$ ratio into a final $XCH_4$ quantity, a model-based estimate of $XCO_2$ is used, according to the equation:

$$XCH_{4(proxy)} = \frac{XCH_4}{XCO_2} \times XCO_{2(model)} \tag{1}$$

where the relative variability of $CO_2$ in the atmosphere is known to be much lower than that of $CH_4$. This leads to the primary disadvantage of this method, i.e. that the model-based estimates of $XCO_2$ may introduce biases in the retrieved $CH_4$. In an attempt to minimise such biases, $CO_2$ dry-air mole fractions $XCO_{2(model)}$ used in Eq. (1) in the UoL proxy retrieval scheme are obtained by taking the median of the estimates produced by three atmospheric chemistry transport models which have assimilated surface in-situ data: GEOS-Chem (Feng et al., 2011), NOAA CarbonTracker (Peters et al., 2007), and CAMS (Chevallier et al., 2010).

The advantage of the Proxy retrieval approach compared to the "Full-Physics" retrieval as typically used for $CO_2$ (Boesch et al., 2011; Cogan et al., 2012), is that Proxy retrievals are less sensitive to instrumental effects, and require less-strict quality filtering (Schepers et al., 2012; Parker et al., 2015), thereby ensuring a better coverage of regions (especially in the tropics), where Full-Physics retrievals are particularly challenging.

## 4.1 Retrieval Inputs and A Priori Generation

In order to prepare all of the necessary inputs to the retrieval, we use the Leicester Retrieval Preparation Toolset (LRPT) software. The latest version of the GOSAT Level 1B files (version 210.210) are acquired directly from the NIES GDAS Data Server and are processed with the LRPT to extract the measured radiances along with all required sounding-specific ancillary information such as the measurement time, location and geometry. These measured radiances have the recommended radiometric calibration and degradation corrections applied as per Yoshida et al. (2013) with an estimate of the spectral noise derived from the standard deviation of the out-of-band signal. We then format the spectral data for input into the UoL-FP retrieval algorithm and generate a list containing all of the ancillary data necessary to create the retrieval a priori information.

Sounding-specific a priori information are generated for all individual soundings present in the sounding-selector list above. Atmospheric temperature and water vapour profiles are taken from ECMWF ERA-Interim up to August 2019 and ERA-5 thereafter. $CO_2$ profile information is taken from the 16r1 CAMS atmospheric inversion (Chevallier, 2019) and incremented by the NOAA estimated global growth rate for recent years. $CH_4$ profiles are taken from a combination of the MACC-II $CH_4$ inversion (v10-S1NOAA - https://apps.ecmwf.int/datasets/data/macc-ghg-inversions/) for the troposphere and from a dedicated

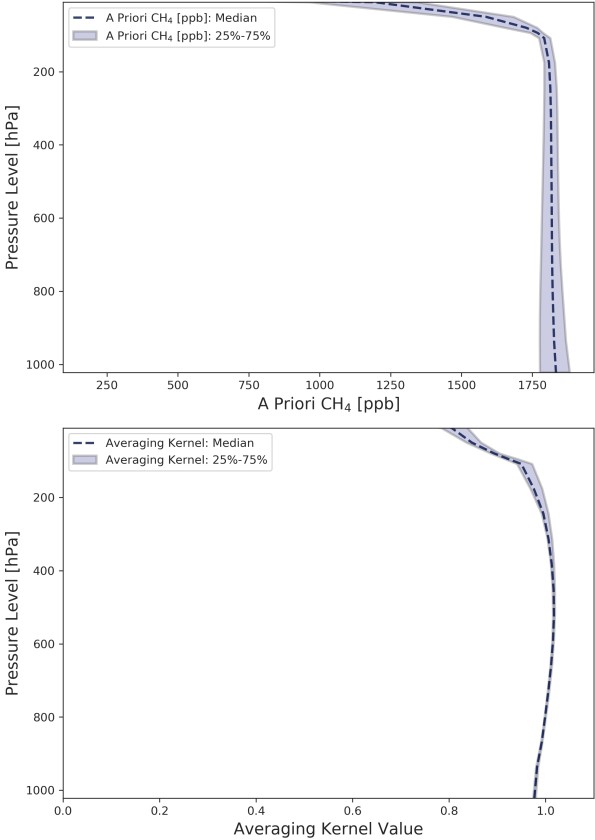

**Figure 1.** The average CH$_4$ a priori profile used in the retrievals along with the 25-75% variation (top). The average normalised column averaging kernel produced by the retrieval along with the 25-75% variation, highlighting the strong sensitivity of the retrieval to the surface (bottom).

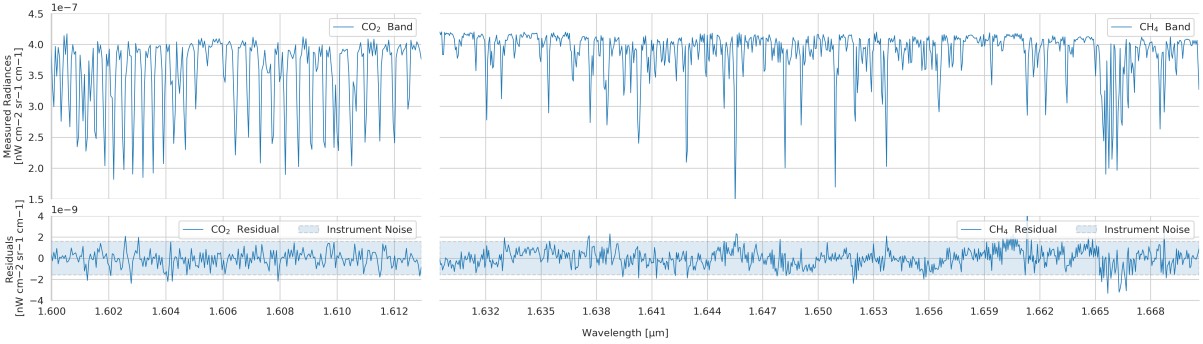

**Figure 2.** Spectra showing the GOSAT radiances for the CO$_2$ and CH$_4$ bands and the resulting residuals (measured - simulated spectral differences). The data shown are the median values for the 25,274 land retrievals that pass the quality filtering for an example month, August 2016.

TOMCAT stratospheric chemistry model simulation (Chipperfield, 1999) for the stratospheric component. This ensures that the $CH_4$ a priori profiles are sufficiently vertically resolved and capture the sharp decrease in concentration around the tropopause (Figure 1 (top)). As the MACC-II data is only available until 2012, the data after this period is repeated each year.

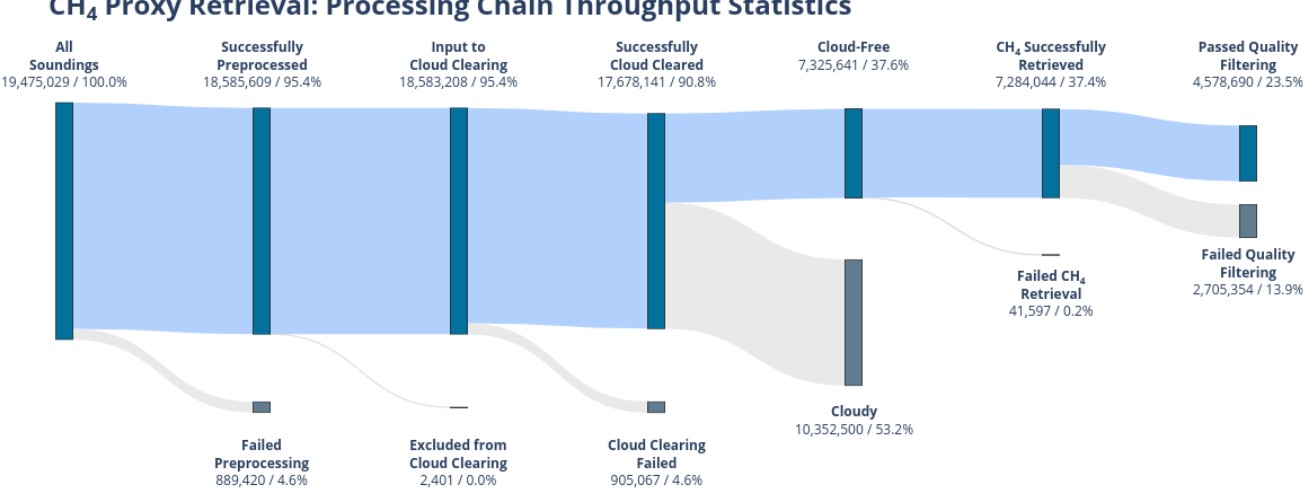

**Figure 3.** Sankey diagram detailing the retrieval throughput at each step of the processing chain for the GOSAT Proxy XCH4 retrieval. As well as the absolute number of soundings, the percentage relative to the initial total is also given.

All of the atmospheric profiles are then interpolated to a sounding-specific retrieval grid. For each sounding a 20-level pressure-based retrieval grid is generated that ranges from the top of the atmosphere (0.1 hPa) to 20 hPa beneath the surface pressure as estimated by ERA-Interim. This 20 hPa buffer allows the surface pressure to be adjusted during the initial cloud-screening process without leading to unphysical extrapolation of the a priori profiles.

In addition to atmospheric a priori information, we also generate sounding-specific a priori information for the spectral dispersion and surface albedo directly from the GOSAT spectra.

The retrieval algorithm also requires the input of spectroscopic parameters for the species being simulated. We use v4.2.0 of the OCO line lists for $CO_2$, $H_2O$ and $O_2$, and take $CH_4$ parameters from the TCCON line lists (Toon, 2015).

## 4.2 Cloud-Screening

Prior to the XCH4 and XCO2 retrievals, cloudy GOSAT soundings are identified and excluded by using the UoL-FP retrieval algorithm to obtain the apparent surface pressure from $O_2$ A-band spectra and comparing it to the surface pressure provided by the ECMWF reanalysis (Dee et al., 2011). If the absolute difference between the retrieved and the ECMWF surface pressure is larger than 30 hPa, a sounding is flagged as cloudy and excluded from further processing. The reason why a loose threshold is used for the surface pressure difference is that this procedure only aims to identify and remove soundings which are significantly

cloudy. Partially cloudy scenes, or scenes where optically thin clouds are present, are processed by the retrieval algorithm, and are dealt with through a post-retrieval quality filtering scheme described later in this section.

## 4.3 XCH$_4$ and XCO$_2$ Retrievals

For soundings that pass the cloud screening procedure described above, retrievals with the UoL-FP algorithm for CO$_2$ and CH$_4$ mole fraction profiles are carried out separately. The state vector for these retrievals consists of 20-level profiles for CH$_4$ and CO$_2$ mole fractions along with profile scaling factors for H$_2$O mole fraction and temperature with parameters for surface albedo and spectral dispersion also included allowing us to explicitly fit the wavelength for each spectra independently.

A post-retrieval quality filtering is then carried out, by selecting the retrievals that meet the following criteria: (1) goodness-of-fit ($\chi^2$) parameter between 0.4 and 1.9 for both CH$_4$ and CO$_2$; (2) a posteriori error smaller than 20 ppb for CH$_4$ and 3 ppm for CO$_2$; (3) retrieved XCH$_4$ larger than 1650 ppb and XCO$_2$ larger than 350 ppm; and (4) latitude north of 60°S (to exclude Antarctica).

Figure 2 shows an example of the spectral fits for one month of data (August 2016) for all 25,274 successful, quality-filtered data measured over land. The top panels show the averaged measured radiances in the 1.6 μm CO$_2$ (left) and 1.65 μm CH$_4$ (right) retrieval windows, with the bottom panels showing the residual differences to the final simulated spectra. The estimated instrument noise is indicated by the shaded area and the residuals are found to be within the noise.

The retrieved XCH$_4$ and XCO$_2$ satisfying the aforementioned quality criteria are then used in Eq. (1), together with the ensemble median model XCO$_2$ described earlier in this section. Prior to the calculation of XCO$_2$ to be used in Eq. (1), model CO$_2$ profiles are convolved with scene-dependent instrument averaging kernels computed as part of the CO$_2$ retrieval.

Before the final production of the data files, an offset is subtracted from the retrieved XCH$_4$ to remove a residual mean bias to TCCON (see Section 5). Currently, a single offset value of 9.06 ppb is used. This offset is applied to all analysis presented here and is built-in to the final delivered data.

A summary of the throughput of the whole processing chain described in this section is shown in Fig. 3. This shows that in total, between April 2009 and December 2019 we have 19.5 million individual GOSAT soundings which are ingested into the LRPT software. Of these, 95.4% are successfully preprocessed, with the 4.6% that fail largely due to incomplete or invalid L1B data. A very small number of successfully preprocessed soundings < 0.1% are unable to be processed further as we are unable to estimate a noise for those spectra. Of the 18.6 million soundings that continue and are attempted for cloud-clearing, 17.7 million are able to be successfully cloud-cleared. Of these, just over 7 million soundings are found to be cloud free, with over 10 million determined to be cloudy. A successful CH$_4$ retrieval is performed on the majority of these cloud-free soundings, with just 41,597 failing the retrieval. Of the 7.3 million successful CH$_4$ retrievals, 2.7 million are rejected by our final quality filtering. It should be noted that currently we exclude all retrievals below 60°S (i.e. Antarctica) due to low signal-to-noise and difficulty in distinguishing low cloud from the snow-covered surface. This alone accounts for over 1 million of the 2.7 million rejected retrievals. Finally, we are left with almost 4.6 million successful and quality-controlled XCH$_4$ retrievals (23.5% of the total measurements performed).

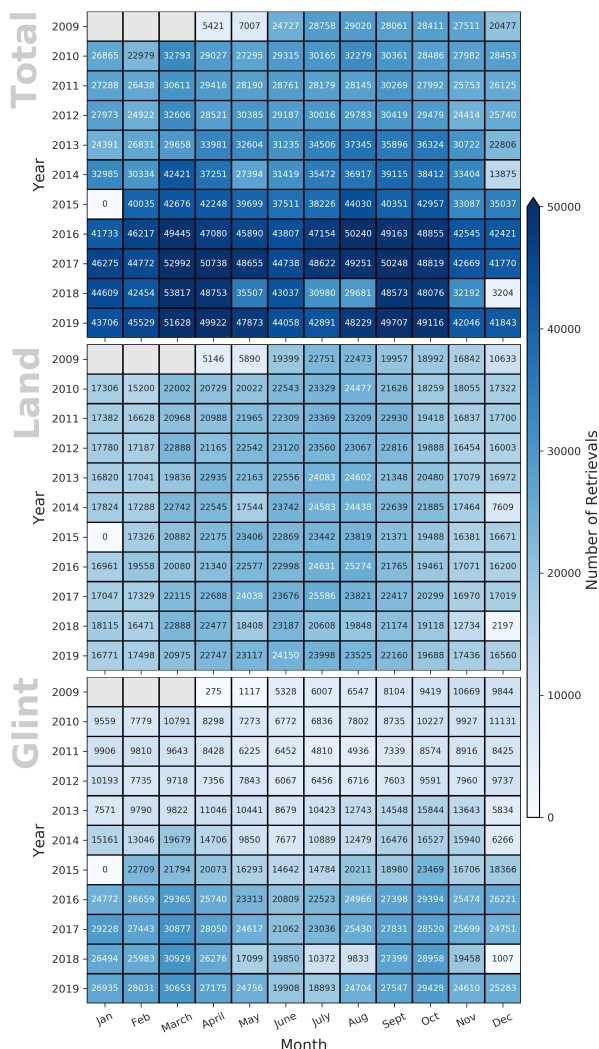

**Figure 4.** The number of GOSAT soundings per month in the final Proxy XCH4 dataset, starting in April 2009 and ending in December 2019. Counts are provided for the global total but also for land and glint observation modes separately.

Figure 4 shows the number of successful retrievals broken down by month and also split into land and glint observation modes. This figure is particularly useful as it highlights any systematic differences in data density over time (e.g. from changes in the GOSAT sampling strategy) and also highlights abrupt data gaps (e.g. from instrument anomalies). In particular, it shows that the increase in monthly data from 2014/2015 onwards is largely a result of an increase in the number of glint observations which is a direct result of the instrument sampling changes that increased the valid glint range. This does highlight that some care should be taken when using the data for some applications as there cannot be assumed consistent temporal/spatial data coverage over the whole data record. Large data gaps, such as in January 2015 and December 2018 are also highlighted and indicate where care may need to be taken when analysing over these periods.

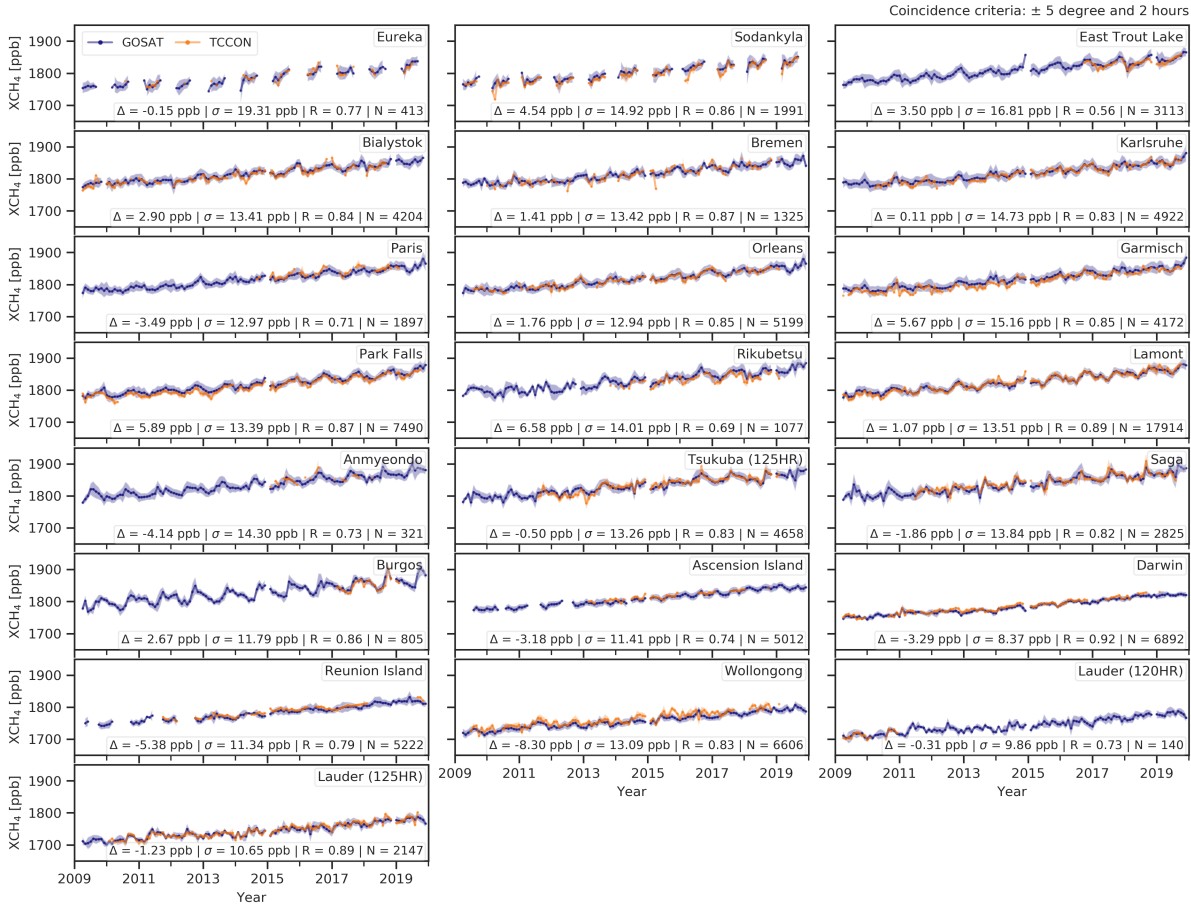

**Figure 5.** Time series plots for each TCCON site comparing GOSAT Proxy XCH$_4$ data to the matching co-located TCCON measurements. Statistics for each site are included, showing the average GOSAT-TCCON difference, the GOSAT-TCCON standard deviation, the correlation coefficients and the number of co-located GOSAT-TCCON measurements.

## 5    Validation Against TCCON

Evaluation against the Total Carbon Column Observing Network (TCCON) is the primary mechanism by which satellite-based measurements of XCO$_2$ and XCH$_4$ are validated.

The TCCON network consists of ground-based high-resolution Fourier transform spectrometers, performing direct measurements of solar spectra in the near-infrared. There are currently 27 operational sites located across North America, Europe, Asia and Oceania, including several islands in the southern hemisphere and other remote areas. TCCON sites have become operational at different times (see Table A1) and hence the data record length varies between sites, with Burgos (Phillipines) (Velazco et al., 2017) and Nicosia (Cyprus) the most recent to come online in 2017 and 2019 respectively. Also note that the Lauder site (Pollard et al., 2017) has multiple instruments and we have kept these records separate.

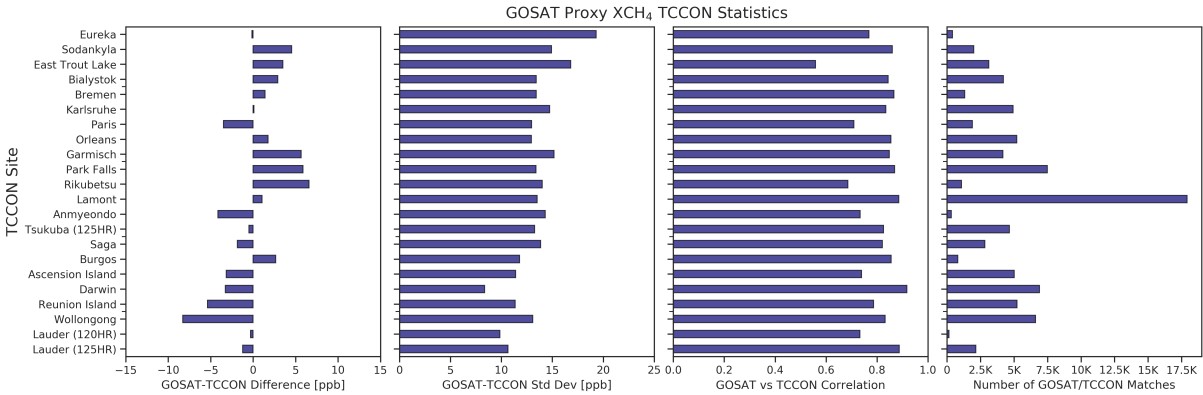

**Figure 6.** Summary of the statistics from Figure 5 comparing the GOSAT Proxy XCH₄ data to the TCCON measurements for each TCCON site. Panels show the GOSAT-TCCON difference, the GOSAT-TCCON standard deviation, the correlation coefficients between the GOSAT and TCCON data and finally the number of co-located GOSAT-TCCON measurements.

TCCON has been used extensively to validate satellite observations of XCH₄ from SCIAMACHY (De Mazière et al., 2004; Dils et al., 2006), GOSAT (Dils et al., 2014; Parker et al., 2011; Yoshida et al., 2013; Parker et al., 2015; Inoue et al., 2016) and TROPOMI (Hu et al., 2018; Schneising et al., 2019; Lambert et al., 2019) and allows these measurements to be bias-corrected where necessary. TCCON itself is tied to the World Meteorological Organization (WMO) standard through comparison against

integrated aircraft measurements (Wunch et al., 2010, 2011).

This work uses the latest available TCCON data, GGG2014. Detailed dataset citations are available for each site in Table A1.

For comparison between TCCON and GOSAT, all GOSAT soundings within $\pm 5°$ of a TCCON site are taken. For these soundings, the average of the TCCON data within $\pm 2$ hours of the GOSAT overpass time is calculated, resulting in GOSAT-

TCCON pairs when there is TCCON data available. It is these matched GOSAT-TCCON pairs that are then subsequently analysed. In total we use 22 of these sites within our analysis (see Table A1), omitting some sites with insufficient data coverage or high-altitude sites where the total column may not be well-represented or co-located well to the satellite observations. Figure 5 shows the time series of the GOSAT (blue) and TCCON (orange) data between 2009 and 2019 for each individual site. Also provided are the mean GOSAT-TCCON difference ($\Delta$), the standard deviation of the GOSAT-TCCON difference

($\sigma$), the correlation coefficient ($R$) and the total number of GOSAT-TCCON pairs (N). These statistics are also summarised in Figure 6 which show that generally the GOSAT-TCCON difference is small (< 5 ppb), the standard deviation (which can be considered to be the single measurement precision of the GOSAT data) is typically between 10-15 ppb, the correlation coefficient is generally high (0.7-0.9) and there are many co-located GOSAT-TCCON measurements with the distribution changing considerably between TCCON sites. Another important validation metric is the relative accuracy, or inter-station

bias. This metric is an indication of any spatio-temporal variability of the bias and is defined in Dils et al. (2014) as the standard deviation of the individual site biases. We obtain a value of 3.89 ppb for this metric, again smaller than the estimated

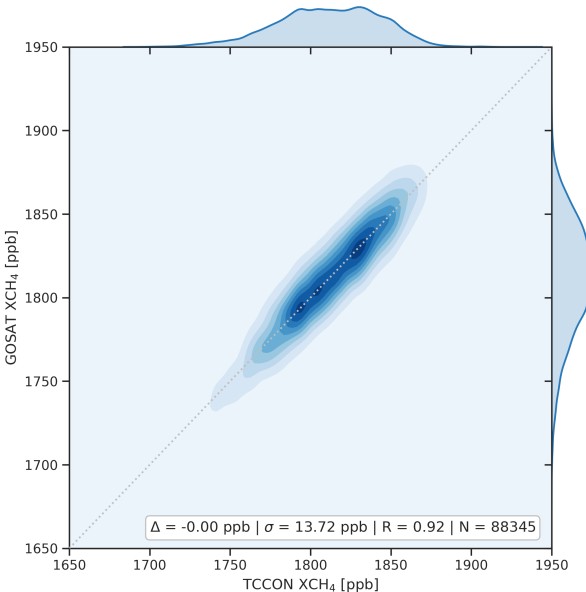

**Figure 7.** Correlation between the 88,345 matching TCCON XCH₄ data and co-located GOSAT Proxy XCH₄ measurements across all TCCON sites. The data is presented as a 2-dimension kernel density estimation (KDE) plot. The distribution sites along the one-to-one line (grey dashed) with a standard deviation (i.e. single-sounding precision) of 13.68 ppb and a correlation coefficient (r) of 0.91. An overall bias to TCCON of 9.06 ppb is removed from the GOSAT data, resulting in an average bias of 0 ppb by design. The individual KDE plots are shown along the upper and right margins.

TCCON accuracy of $\pm 4$ ppb. This meets the "breakthrough" user requirement for the systematic error of 5 ppb as defined by Buchwitz et al. (2017a).

In total across all TCCON sites we find 88,345 matching GOSAT-TCCON data pairs. The correlation between the GOSAT and TCCON data is shown in Figure 7, presented as a 2-D kernel density estimation (KDE) plot, along with the corresponding marginal 1-D KDE plots on the X and Y axes. An overall difference of 9.06 ppb is removed from the GOSAT data so that, by design, the absolute average difference to TCCON is 0 ppb. The overall standard deviation or single measurement precision is found to be 13.72 ppb with a correlation coefficient of 0.92. The single measurement precision of 13.72 ppb comfortably exceeds the precision breakthrough requirement of 17 ppb (Buchwitz et al., 2017a) indicating that it "would result in a significant improvement for the targeted application". Although the data contributing to this plot are from a wide variety of TCCON sites in different locations and at different latitudes, the distribution appears consistent and is tightly aligned to the one-to-one (dashed) line. However, there are signs of a potential hemispheric or latitudinal bias in the data against TCCON, although this is not apparent at all sites; for example Karlsruhe, Lamont, Tsukuba and Lauder all have negligible biases but span a large latitude range. It should also be noted that the uncertainty on the TCCON XCH₄ is approximately 4 ppb and for the majority of sites the GOSAT-TCCON difference is within this uncertainty so care must be taken to not over-interpret any signals at this scale.

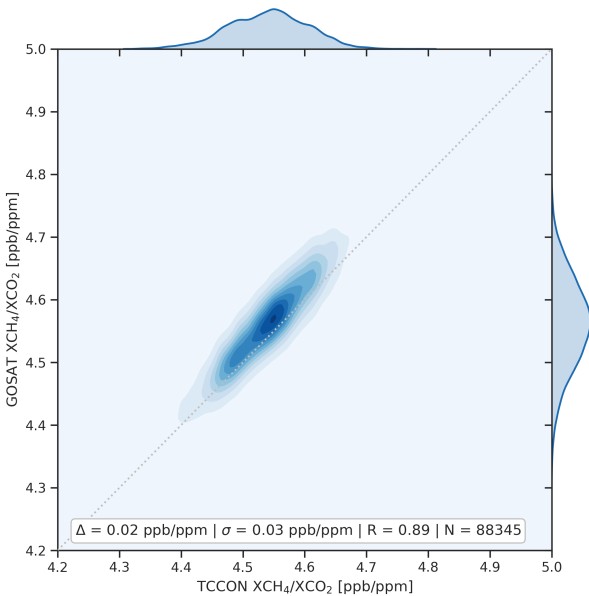

**Figure 8.** Correlation between the 88,345 matching TCCON $XCH_4/XCO_2$ ratios and co-located GOSAT XCH/$XCO_2$ ratios retrieved as a raw fundamental part of the Proxy $XCH_4$ retrieval (Equation 1). The data is presented as a 2-dimension kernel density estimation (KDE) plot. The distribution sites along the one-to-one line (grey dashed) with a standard deviation of 0.03 ppb/ppm and a correlation coefficient (r) of 0.89. An overall bias to TCCON of 0.02 ppb/ppm is present in this raw data. The individual KDE plots are shown along the upper and right margins.

## 5.1 XCH₄/XCO₂ Ratio Validation

In addition to validation of our final $XCH_4$ Proxy dataset, TCCON data allow us the opportunity to validate the different components in Equation 1, namely the $XCH_4/XCO_2$ ratio and the model-derived $XCO_2$.

Figure 8 shows the correlation between the retrieved GOSAT $XCH_4/XCO_2$ ratio (ppb/ppm) (with no bias correction applied)

5 and the corresponding ratio calculated from TCCON. There is an excellent correlation coefficient of 0.89 across the 88,345 matching data points with a standard deviation of just 0.03 ppb/ppm. An average offset between the two datasets of 0.02 ppb/ppm exists and is of a very similar magnitude to the global offset that is removed from the final data of 9.06 ppb. To be clear, the final bias correction which we apply to the Proxy $XCH_4$ is almost entirely attributed to this bias that we identify here in the $XCH_4/XCO_2$ ratio. It should also be noted here that the TCCON data itself has a bias correction applied to the

10 $XCO_2$ and $XCH_4$ data. This airmass-independent correction factor derived from airborne calibrations is 1/0.9898 for $XCO_2$ and 1/0.9765 for $XCH_4$ (Wunch et al. (2010) - Table 5). It is considered that this correction is mainly a result of deficiencies in the spectroscopy, which likely apply to the GOSAT retrievals as well and might go some way to explaining this small difference between TCCON and GOSAT.

## 5.2 Validation of XCO$_2$ Model

To validate the XCO$_2$ model data used in the generation of the final Proxy data, we evaluate the model median XCO$_2$ mixing ratios against TCCON but also evaluate the three individual models, sampled at the time and location of the GOSAT soundings, with the GOSAT sounding-specific averaging kernel applied. These XCO$_2$ models are all independent of TCCON data but do assimilate NOAA surface site measurements, some of which are nearby to TCCON sites. Figure 9 shows the correlation between TCCON and the Model Median XCO$_2$ (top left), GEOS-Chem XCO$_2$ (top right), CAMS XCO$_2$ (bottom left) and CarbonTracker XCO$_2$ (bottom right). In all four cases there is an excellent agreement between TCCON and the model data with correlation coefficients all at 0.99, with average differences ranging from -0.07 ppm to 0.17 ppm and standard deviations between 1.02 ppm to 1.17 ppm. Although the values are all very similar, the model with the smallest difference (GEOS-Chem) has the largest standard deviation and conversely the model with the largest difference (CAMS) has the smallest standard deviation. Overall the Model Median performs marginally better than any of the individual models with a standard deviation of 1.02 ppm but retaining a very small difference (0.08 ppm). Parker et al. (2015) provides a detailed assessment of the contribution to the overall uncertainty on the Proxy XCH$_4$ related to the model XCO$_2$.

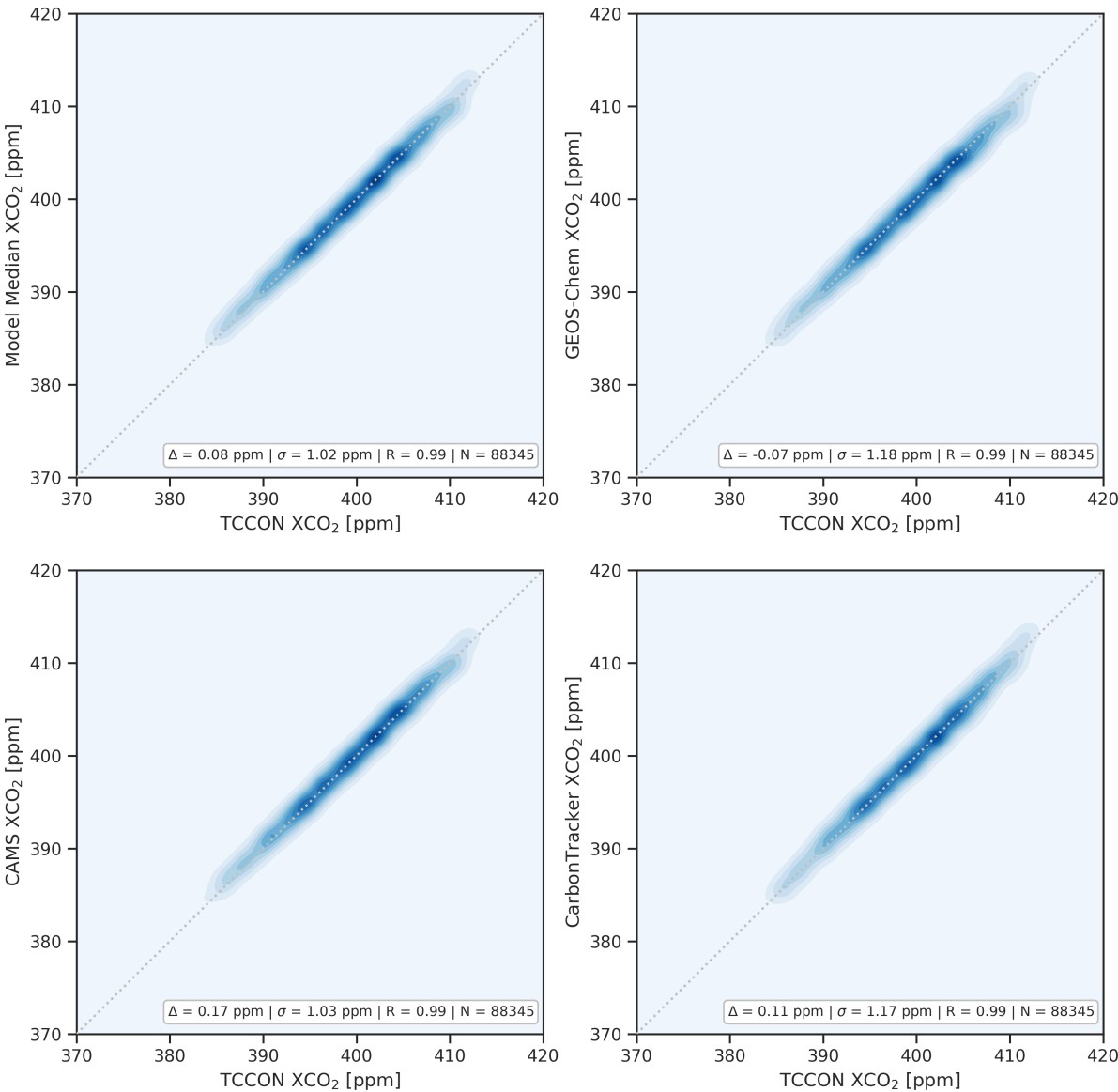

**Figure 9.** Correlation of the individual XCO$_2$ models (GEOS-Chem, MACC and Carbon-Tracker) used in the generation of the GOSAT Proxy XCH$_4$ data against matching co-located TCCON XCO$_2$ measurements. The first panel shows the median model value which is the quantity directly used in Equation 1 to generate the final Proxy XCH$_4$ quantities. The data is presented as a 2-dimension kernel density estimation (KDE) plot with the one-to-one line shown as the grey dashed line. Also included are the statistics (difference, standard deviation, correlation coefficient and number of matching measurements) comparing each model to the TCCON XCO$_2$.

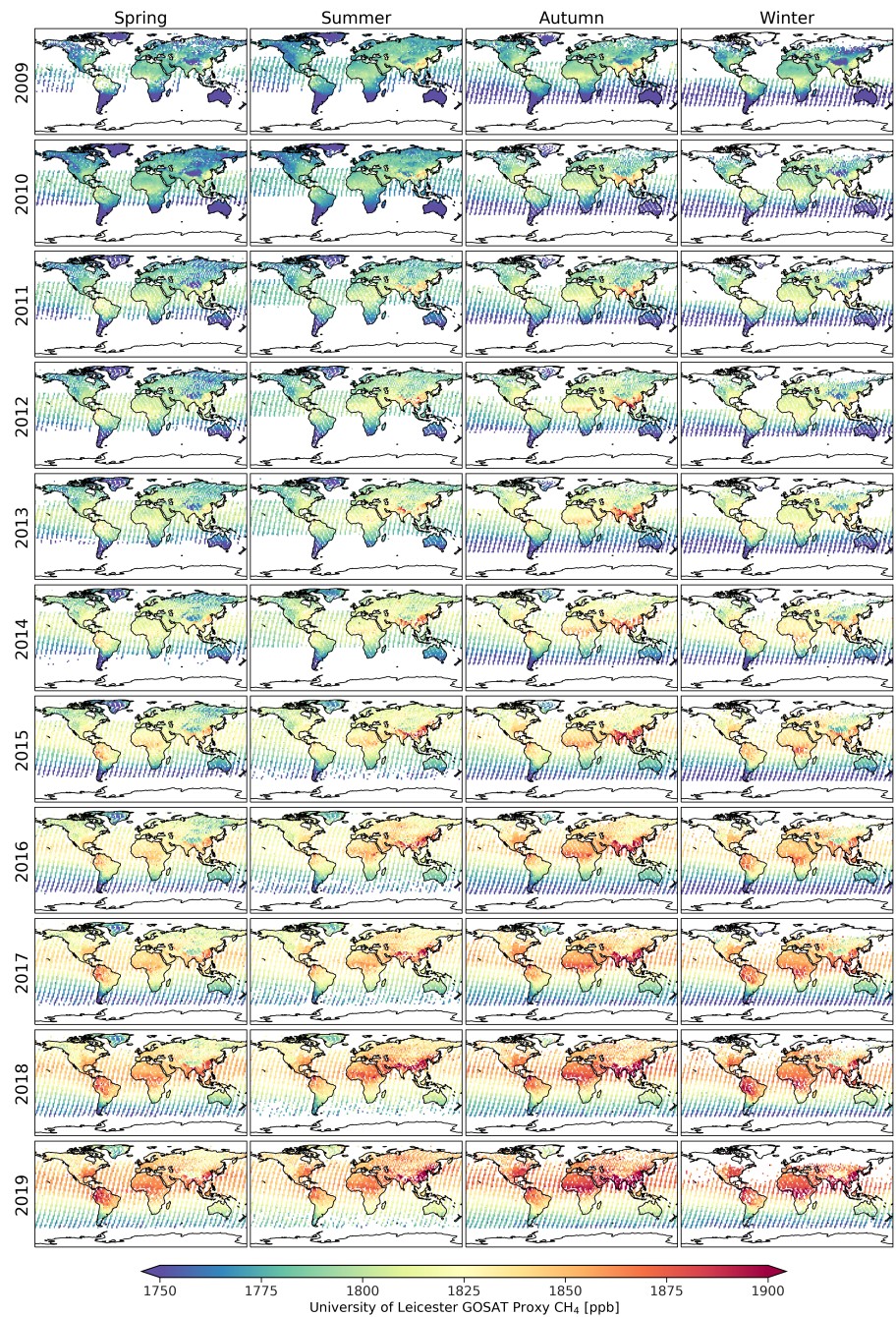

**Figure 10.** Global maps showing the GOSAT Proxy XCH$_4$ data from April 2009 to December 2019 separated into seasons - Spring (MAM), Summer (JJA), Autumn (SON) and Winter (DJF).

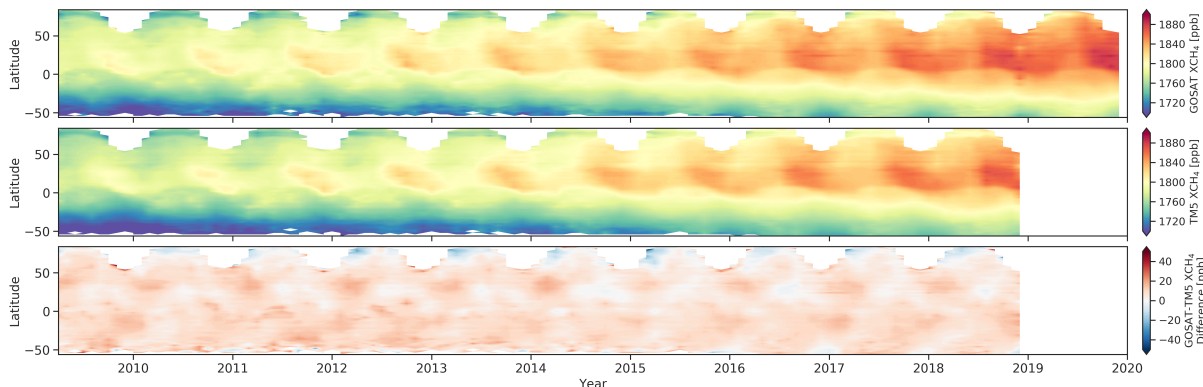

**Figure 11.** Hovmöller (Latitude vs Time) plot of the GOSAT Proxy XCH$_4$ (top), TM5 model XCH$_4$ (middle) and the GOSAT-TM5 difference (bottom). The model data has been sampled at the time/location of the GOSAT observation and had the sounding-specific GOSAT averaging kernel applied.

## 6   Global CH$_4$ Distributions

As discussed in Section 3, one of the primary applications for the GOSAT Proxy XCH$_4$ data has been as input to global flux inversions. For this reason, it is useful to examine the global spatio-temporal distribution of the data. Figure 10 shows seasonal maps of the GOSAT data from Spring (March/April/May) 2009 through to Winter (December) 2019. Features of note include: a consistent increase in concentration over time; strong regional signals associated with CH$_4$ surface sources, particularly over South America, India, China and Africa; a clear seasonal cycle over many regions; and a significant increase in the number and latitudinal range of GOSAT ocean sun-glint observations from 2014/2015.

Despite changes in the GOSAT sampling pattern and various instrument issues resulting in data gaps (see Section 2.1), on a seasonal and global scale there is good data coverage throughout the entire decade of observations.

## 7   Model Comparisons

The purpose of this paper is to present details of the v9.0 Proxy dataset and provide information to facilitate the future use of the data. As such, it is not the intention that this work performs detailed scientific analysis and interpretation. We do not, for example perform any atmospheric flux inversions using this data as that is a significant study in its own right. However, we do feel that it would be informative to users of the data for us to perform a comparison against existing model XCH$_4$ simulations to give confidence that the data is of sufficient quality to use in such studies.

In this section we compare the GOSAT Proxy XCH$_4$ data to a simulation of the TM5 global chemistry transport model (Bergamaschi et al., 2013, 2018b) which has assimilated NOAA surface measurements. We have chosen to compare against model simulations that are both widely used within the community and that have already assimilated NOAA surface measurements. The reasoning for this is that any overall differences as might be seen from free-running model simulation are removed and we can clearly state the consistency of our dataset with the NOAA network. By proving good overall agreement to both

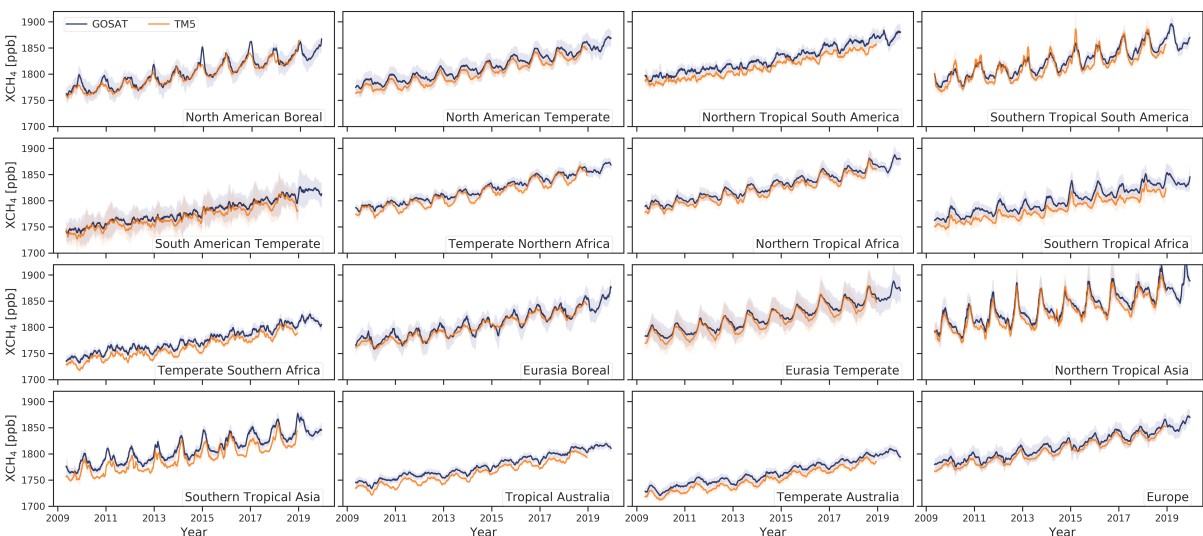

**Figure 12.** Time series comparison between the GOSAT Proxy $XCH_4$ data and TM5 model $XCH_4$ simulations (Bergamaschi et al., 2018b) for individual TransCom regions. The model data has been sampled at the time/location of the GOSAT observation and had the sounding-specific GOSAT averaging kernel applied.

TCCON measurements (Section 5) and the NOAA-constrained model simulations we believe this indicates the utility of our data for use in further scientific analysis. This model data is the same data as used in the *Global Methane Budget 2000-2017* (Saunois et al., 2020) and in that study is referred to as "TM5-4DVAR". For these comparisons, we have sampled the TM5 model at the time and location of the GOSAT measurement, interpolated the model to the GOSAT retrieval grid, applied the sounding-specific GOSAT $CH_4$ averaging kernel (see Figure 1 and Appendix D) and computed the model total column $XCH_4$ amount.

Figure 11 shows Hovmöller plots (latitudinal mean versus time) for the GOSAT Proxy $XCH_4$ (top), TM5 model simulation (middle) and the GOSAT-TM5 difference (bottom). The GOSAT distribution behaves as expected, showing an increase over time between 2009 and 2019 superimposed on top of a north-south gradient and regular seasonal cycle. The TM5 data exhibits very similar characteristics and is in very good agreement to the GOSAT data. The difference between the two datasets (lower panel) shows that although there is a small offset between the two (with GOSAT on average 6.55 ppb larger than the model), there is very good consistency over time. GOSAT and TM5 seem to agree slightly better during the peak of the seasonal cycle, particularly in the tropics, with the TM5 exhibiting a shallower trough. At very high northern latitudes, GOSAT is slightly lower than the model but this relates to observations over Greenland at high altitude and low signal to-noise ratio for the GOSAT soundings so care must be taken not to over-interpret this difference.

With Figure 11 providing confidence that the GOSAT data and TM5 simulation are in broad agreement, it is informative to break these comparisons down to a regional scale. Figure 12 shows timeseries of the GOSAT and TM5 data over the 16 different TRANSCOM regions (Gurney et al., 2002). All regions show good agreement between the modelled and observed data. We compute the de-seasonalised $XCH_4$ over time for each region. The average difference in model and observation ranges from

3.9 ppb (Eurasian Boreal) to 15.4 ppb (Southern Tropical Asia). On average across all regions, the mean difference between the model and observation is 9.8 ppb.

The observed seasonal cycles in each region are very well-represented by the model, with an average correlation coefficient of 0.93 (ranging from 0.84 to 0.98 across all regions). The peak to peak seasonal cycle timing and magnitude is very well reproduced between the two datasets. For example, the average peak to peak seasonal cycle amplitude for Northern Tropical Asia is 61.1 ppb for GOSAT and 62.3 ppb for TM5 whilst for North American Temperate it is 22.7 ppb for GOSAT and 23.6 ppb for TM5. The average GOSAT-model difference between the mean seasonal cycle amplitude across all regions is -0.84 ppb with the average absolute difference being 5.3 ppb with only a small number of instances where the model and observation strongly disagree (for example in North America Boreal and South America Tropical in 2015, likely indicative of isolated regional emissions).

It is not the purpose of this paper to diagnose or interpret detailed differences between the observations and model but it is useful to make a few observations relevant for use of the data within future studies. Figure 12 indicates a potential latitude-dependent bias, which is most likely due to model deficiencies in simulating the stratosphere (especially at mid and high latitudes (Patra et al., 2011; Alexe et al., 2015; Saad et al., 2016; Wang et al., 2017)) or inadequate inter-hemispheric mixing, but could partly indicate also some latitudinal bias of the satellite retrievals. Accounting for such a latitudinal dependence through the fitting of a second-order polynomial function (as in Bergamaschi et al. (2013); Turner et al. (2015)) may improve the baseline agreement between model and observation and is an approach that users may wish to explore depending upon their application. Furthermore, these model simulations are constrained by the NOAA background observations. Therefore differences between TM5-4DVAR and GOSAT may partly reflect deficiencies of bottom-up inventories used as prior, particularly over strong emission regions (e.g. obvious deficiencies in tropical Africa related to wetlands). When incorporating the GOSAT data into such inversions, this leads to the production of significant increments in the inverted fluxes and better agreement between observation and simulation (as in Alexe et al. (2015) and other studies noted in Section 3.4).

# 8    Summary and Outlook

In this work we have presented the latest version of the University of Leicester GOSAT Proxy $XCH_4$ dataset. This dataset now contains over a decade of global $CH_4$ observations, sensitive to surface emissions and hence suited to estimating $CH_4$ fluxes. The capability to estimate global and regional $CH_4$ emissions is vital to improving our understanding of the global methane budget and how this budget may respond and change with respect to a changing future climate.

We begin this work by highlighting the wide variety of studies that previous versions of this dataset have contributed towards, demonstrating the significant utility of this dataset for examining and understanding the global methane budget.

This work provides a thorough description of the data processing chain, explaining in detail how the data is generated and how the high-quality of the dataset is ensured. Extensive validation of the data against the TCCON network is performed, validating not only the final Proxy $XCH_4$ data but also the separate components (the $XCH_4/XCO_2$ ratio and the modelled $XCO_2$) that form the final data product.

We also provide global seasonal maps of the data that demonstrate the global distribution of the data as well as highlighting particular features and regions that may be of interest for more detailed study.

Finally, as the primary usage of the data is expected to be as input into a flux inversion data assimilation framework in conjunction with atmospheric chemistry transport models and observations from surface networks, it is useful to compare the consistency against existing model simulations. We compare zonally and regionally against TM5 simulations that have assimilated observations from the NOAA surface network. We find generally a high level of consistency whilst identifying the additional utility that the satellite observations should introduce to the system.

Despite GOSAT-1 having a planned mission lifetime of 5 years, it continues to successfully perform measurements 11 years after launch. GOSAT-2 was launched in October 2018 (Suto et al., 2020) and will continue the legacy of the GOSAT-1 mission. GOSAT-2 offers several opportunities for development related to the dataset we describe here. Primarily, it ensures that should GOSAT-1 cease operation, the valuable decade-long timeseries of observations can continue to be extended via GOSAT-2. With a significant overlap in time between the two missions, consistency between the two missions can be assured, albeit with significant future work/development.

In addition, GOSAT-2 has additional capabilities, namely the possibility of measuring carbon monoxide (CO). By measuring $CO_2$, $CH_4$ and CO simultaneously from the same instrument, GOSAT-2 would allow the extension of studies examining biomass burning combustion, leading to constraints on fire emission ratios as have been performed previously for GOSAT-1 (Ross et al., 2013; Parker et al., 2016).

A strong focus of future $CH_4$-measuring satellites will be to examine anthropogenic emission sources at very high spatial resolution (e.g. PRISMA (Pignatti et al., 2013); HISUI (Matsunaga et al., 2017); ENMAP (Guanter et al., 2015)), particularly relating to monitoring of the oil and gas industry. However, many scientific challenges and questions remain regarding the long-term $CH_4$ behaviour and the response to a changing climate. For this reason, a long-term, consistent climate-ready data record as we present here is of continued importance. We expect that this data will be valuable for numerous studies, from regional flux inversions to monitoring long-term trends. With now over a decade of global atmospheric $XCH_4$ observations, this dataset has helped, and will continue to help, us better understand the global methane budget and investigate how it may respond to a future changing climate.

## 9 Data availability

The University of Leicester GOSAT Proxy v9.0 $XCH_4$ data (Parker and Boesch, 2020) is available from the Centre for Environmental Data Analysis data repository at https://doi.org/10.5285/18ef8247f52a4cb6a14013f8235cc1eb. The TCCON data is available from the TCCON Data Archive at https://tccondata.org (individual data citations are provided in Table A1). CAMS model $CO_2$ (v18r2) data is available from the Copernicus Atmospheric Data Store at https://ads.atmosphere.copernicus.eu/cdsapp#!/dataset/cams-global-greenhouse-gas-inversion. MACC model $CH_4$ (v10-S1NOAA) is available from ECMWF at https://apps.ecmwf.int/datasets/data/macc-ghg-inversions/. NOAA CarbonTracker model $CO_2$ (CT2017 and CT2019-NRT) are available from NOAA ESRL, Boulder, Colorado, USA at ftp://aftp.cmdl.noaa.gov/products/carbontracker/co2/.

## Appendix A: Summary of Dataset Characteristics

Table A1 summarises the key characteristics of the University of Leicester GOSAT Proxy XCH$_4$ data, including the spatial and temporal extent that the dataset covers, the total number of measurements and their evaluation against TCCON.

| Attribute | Value |
|---|---|
| Temporal Extent | 2009-2019 |
| Spatial Extent | Global (56.3°S - 83.5°N) |
| Total Number of Measurements | 4.6 million |
| Footprint Size | 10.5km (at nadir) |
| Overpass Time (at Equator) | ~13:00 Local Solar Time |
| Bias (vs TCCON) | 0 ppb (after global bias correction of 9.06 ppb) |
| Precision (vs TCCON) | 13.72 ppb |

**Table A1.** Table summarising the key characteristics of the University of Leicester GOSAT Proxy XCH$_4$ data.

## Appendix B: Previous Data Versions

Table B1 outlines the history and evolution of the University of Leicester GOSAT Proxy XCH$_4$ data product. Entries include the version number, the project that the data was generated for, the version of the GOSAT L1B data used, the time period covered by the data, whether ocean sun-glint data was generated, comments relating to changes/updates from previous versions and peer-reviewed publications that we are aware of that used the data. For the ESA GHG-CCI project, we also indicate which versions were officially delivered as part of the Climate Research Data Packages through the project. All Copernicus C3S

versions were delivered to the Copernicus Climate Data Store.

## Appendix C: Data Contents and Usage Notes

This section provides information on the contents and usage of the netCDF data files that we provide containing the Proxy XCH$_4$ data. Whilst we recommend that anyone using the data should discuss their specific usage with the author, the following information is useful to note.

Our data is delivered as daily netCDF files, containing `n` individual GOSAT soundings. We provide everything in the data files that we believe users would require to make use of our data, including our a priori information (`ch4_profile_apriori`) and averaging kernels (`xch4_averaging_kernel`) which are provided on `m` vertical levels (see Appendix D).

In general, users should **only use data that passes our quality checks (i.e. `xch4_quality_flag == 0`)**. In some specific use cases, the data that has failed our checks may still be of use but additional care should be taken in using this data and we

strongly recommend discussing such applications with us to determine if that is suitable for your use.

| Version | Project | L1B | Time Period | Ocean? | Comments | Publications Utilising Data |
|---|---|---|---|---|---|---|
| 1.0 | NCEO | 006 007 | 2009-2010 | No | First release. | Parker et al. (2011) |
| 2.0 | NCEO | 050 080 100 | 2009-2010 | No | Development version; only processed at TCCON overpasses; new radiometric calibration for L1B from JAXA applied. | |
| 3.0 | CCI | 050 080 100 | 2009-2010 | No | First version generated as part of ESA CCI Round Robin and Algorithm Intercomparison. Improvements to surface pressure calculation taking into account instrument field-of-view. Significant speed improvements to preprocessing steps to allow multi-year processing. | |
| 3.1 | CCI | 050 080 100 | 2009-2010 | No | Incremental updates and bug fixes. | |
| 3.2 | CCI | 050 080 100 | 2009-2010 | No | Incremental updates and bug fixes. First version to be used for atmospheric inversions. | Ross et al. (2013); Fraser et al. (2013); Dils et al. (2014); Wecht et al. (2014); Cressot et al. (2014); Berchet et al. (2015); Worden et al. (2015) |
| 4.0 | CCI CRDP1 | 141 150 151 | 2009-2011 | No | First version that was widely released to scientific community via CCI. Change in $CO_2$ model from CarbonTracker to MACCII; tightening of stratospheric covariance; spectral degradation applied as per Yoshida et al. (2013); higher resolution ERA-Interim; variable vertical grid based on tropopause height. | Fraser et al. (2014); Turner et al. (2015); Alexe et al. (2015) |
| 5.0 | CCI | 160 161 | 2009-2013 | No | Introduction of model median $XCO_2$ with MACCII, CarbonTracker and GEOS-Chem (as per Parker at al, 2015); $CH_4$ prior combines MACCII (troposphere) and TOMCAT (stratosphere); SRTM DEM used for topography; updated GOSAT L1B degradation correction based on Kuze et al. (2014). | |
| 5.1 | CCI | 160 161 | 2009-2013 | No | Resolved minor issue with uncertainty variable in 5.0. | |
| 5.2 | CCI CRDP2 | 160 161 | 2009-2013 | No | Resolved minor issue with pressure weighting function In 5.1. | Parker et al. (2015); Stanevich et al. (2019, 2020) |
| 6.0 | CCI CRDP3 | 160 161 | 2009-2014 | Yes | Inclusion of ocean sun-glint observations for first time. | Webb et al. (2016); McNorton et al. (2016b); Siddans et al. (2017); Feng et al. (2017); Buchwitz et al. (2017b, 2018) |
| 6.1 | CCI | 160 161 | 2009-2015 | Yes | Temporal extension of 6.0. | Ganesan et al. (2017) |
| 7.0 | CCI CRDP4 | 201 202 | 2009-2015 | Yes | New L1B data and updates to $CO_2$ models used in ensemble for Proxy calculation. | Parker et al. (2016, 2018); McNorton et al. (2018); Sheng et al. (2018); Maasakkers et al. (2019, 2020); Lunt et al. (2019) |
| 7.0 | C3S | 201 202 | 2009-2016 | Yes | Time period extended and first delivery to C3S. No changes to processing so retained version number. | |
| 7.1 | C3S | 201 202 210 | 2009-2017 | Yes | Extension of 7.0 with updated L1B for latter years. | |
| 7.2 | C3S | 201 202 210 | 2009-2018 | Yes | Extension of 7.1 with updated L1B for latter years. | Zheng et al. (2019); Reuter et al. (2020); Yin et al. (2020); Parker et al. (2020b); Saunois et al. (2020); Tunnicliffe et al. (2020) |
| 8.0 | C3S | 210 | 2009-2018 | Yes | Internal version for testing/development of new processing pipeline. | |
| 9.0 | C3S | 210 | 2009-2019 | Yes | **The dataset described in this publication.** Fully consistent timeseries. Uses new python-based preprocessing (LRPT). | Parker et al. (2020a); Lu et al. (2020); Zhang et al. (2020) |

**Table B1.** Table showing the evolution of the University of Leicester GOSAT Proxy XCH$_4$ data product. Entries include the version number, the project that the data was generated for, the version of the GOSAT L1B data used, the time period covered by the data, whether ocean sun-glint data was generated, comments relating to changes/updates from previous versions and peer-reviewed publications that we are aware of that used the data. For the ESA GHG-CCI project, we also indicate which versions were officially delivered as part of the Climate Research Data Packages through the project. All Copernicus C3S versions were delivered to the Copernicus Climate Data Store.

We provide data from both observation modes (nadir land and ocean sun-glint) in the same file. While we do not believe that we have any bias between these different modes, for some use cases, users may wish to exclude either of these modes. `retr_flag` provides information on the mode for each sounding (0 = land, 1 = glint).

The variable named `xch4` refers to the final Proxy XCH$_4$ as calculated using Equation 1. **This is the main data product**
**that we provide.** In addition to this, we also provide the other components of Equation 1. `raw_xch4` and `raw_xco2` refer to the directly retrieved XCH$_4$ and XCO$_2$ quantities. These variables should generally not be used but may be useful for certain applications. For example, some users may wish to use the XCH$_4$/XCO$_2$ ratio (i.e. `raw_xch4/raw_xco2`) but replace the model XCO$_2$ that we use (`model_xco2`) with their own modelled XCO$_2$ which may be more appropriate to their particular application or more consistent with their own model transport.

Our retrievals are typically performed on 20 vertical levels, with the first (bottom) level being the surface pressure. However, in a number of instances, especially over high terrain, where the apparent surface pressure from our O$_2$-A band cloud screening is above the bottom two levels of our pressure profile, this can result in only 19 active retrieval levels. This data is still valid but variables with a vertical dimension (`m`) will contain a `fill_value` of $-9999.99$ at the first/lowest value. This value should be checked for and that particular profile should be considered to only have 19 levels, rather than the standard 20.

We identify individual GOSAT soundings by their `exposure_id`. This may be of use when attempting to match our data to other GOSAT data products. This is a numerical identification that matches the GOSAT L1B file which the sounding was extracted from, appended with an additional 3 digits (0-indexed) to identify the number of the sounding within that L1B file. Equation C1 shows the structure of the `exposure_id`. The `exposure_id` of 200908010047044013 0006 was the 7th (006) sounding originating from the GOSAT L1B file named `GOSATTFTS2009080100470440130_1BOB1D210210.01`.

The nomenclature for the GOSAT L1B file includes year (2009), month (08), day (01), hour (00), minute (47), orbit (044), scene (0130). Note that these times are the times for the start of that GOSAT scene and not the exact measurement time. We provide the `time` variable as the measurement start time.

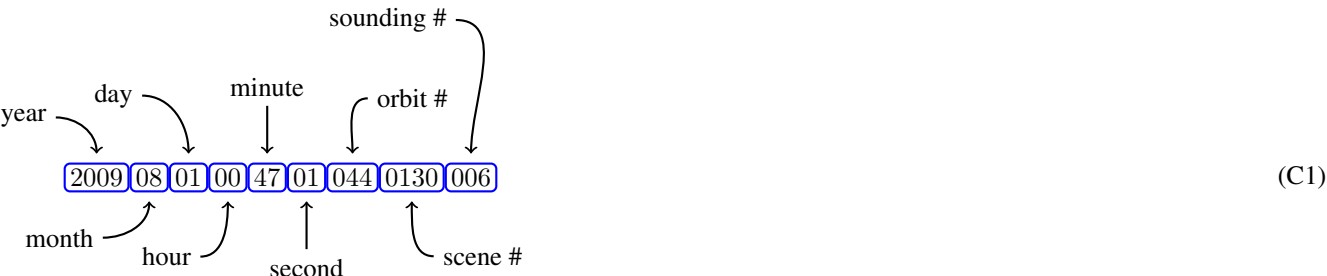

$$\tag{C1}$$

**Appendix D: Application of GOSAT Averaging Kernels to Model Data**

In order to correctly compare any model simulation to the satellite observations, the model data must be transformed to be consistent with assumptions made within the retrieval. Ultimately, this requires the satellite averaging kernels to be applied to the model data with any influence from the a priori data taken into account. The theory and methodology to do this is described in detail in Rodgers (2000) and we only briefly outline the method below. Equation D1 is the equation which should be applied to any $CH_4$ model data and details which variables provided in the data files are required to achieve this. It is assumed that any model data has already been interpolated to the same 20-level pressure grid (`pressure_levels`) as used in the retrieval. It should be noted here that this interpolation should be done with care to try and ensure that the model $XCH_4$ is conserved via the interpolation process. Once on the same vertical grid as the GOSAT a priori and averaging kernels, Equation D1 can be applied to compute the modelled $XCH_4$ by using the `pressure_weight`, `xch4_averaging_kernel` and `ch4_profile_apriori` variables provided in the file. It should be noted that we provide these values for each individual GOSAT sounding and that these are all level (i.e. layer boundary) quantities.

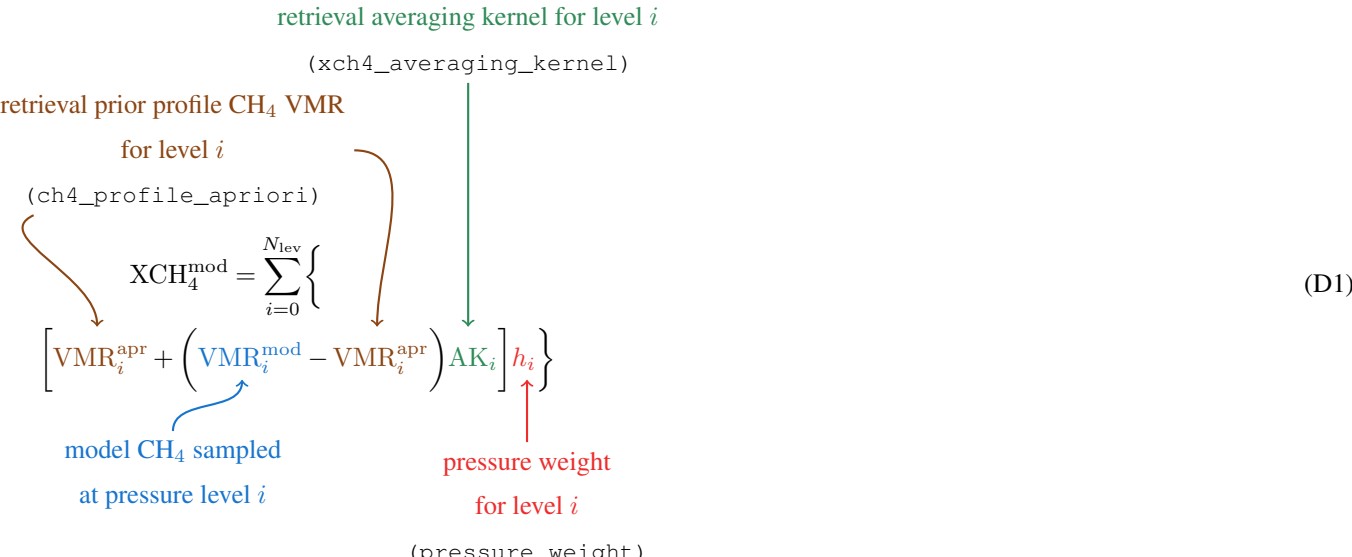

$$\mathrm{XCH_4^{mod}} = \sum_{i=0}^{N_{\mathrm{lev}}} \left\{ \left[ \mathrm{VMR}_i^{\mathrm{apr}} + \left( \mathrm{VMR}_i^{\mathrm{mod}} - \mathrm{VMR}_i^{\mathrm{apr}} \right) \mathrm{AK}_i \right] h_i \right\} \tag{D1}$$

retrieval averaging kernel for level $i$
(`xch4_averaging_kernel`)

retrieval prior profile $CH_4$ VMR
for level $i$
(`ch4_profile_apriori`)

model $CH_4$ sampled
at pressure level $i$

pressure weight
for level $i$
(`pressure_weight`)

**Appendix E: Considerations Regarding GOSAT Measurement Strategy and Changes Over Time**

This section provides details on how the GOSAT measurement strategy has evolved over time. GOSAT initially operated primarily on a regular 5-point grid, revisiting the same grid point. This changed over time to a 3-point grid in order to reduce pointing uncertainty at the extreme angles when using the primary pointing mechanism. However, one consequence of a regular grid is that this resulted in a limited number of observations over some regions, especially islands, where the regular grid point

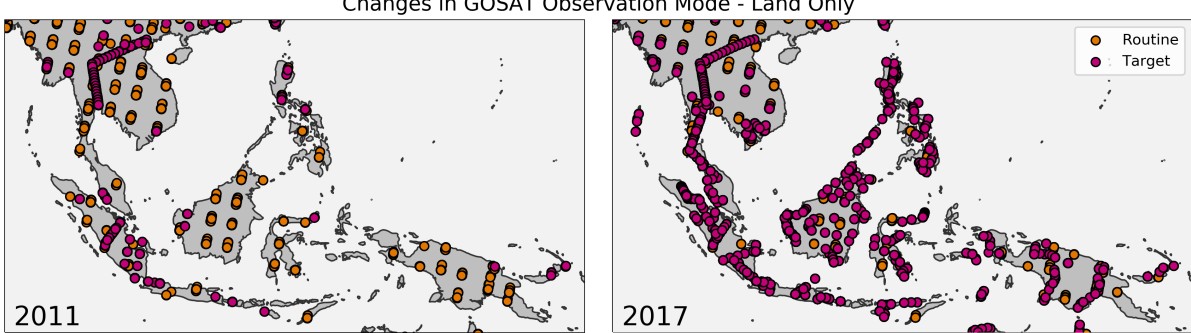

**Figure E1.** Figure contrasting the routine vs target observation modes for land-only measurements in 2011 and 2017 over Indonesia. The change in pointing mechanism has allowed a more sophisticated targeting strategy to optimise the number of land measurements over islands and coastlines.

would fall over the ocean. Thanks to the switch to the secondary pointing mechanism, several changes to the sampling strategy were possible.

Firstly, GOSAT is now capable of targeting more specifically and "follows" coastlines in a more efficient manner. The example in Figure E1 shows the change in sampling location over Indonesia, contrasting 2011 to 2017. Although the exact same grid location is not revisited in the same way, overall there are both more successful measurements and a better geographic coverage.

Secondly, GOSAT is now capable of a wider pointing range and subsequently, the latitudinal range of ocean sun-glint observations have been extended (as observed in Figure 10). Figure E2 provides example comparisons of the measurement density over Australia between the start of the mission (2009-2010) when the instrument was primarily operating in 5-point grid mode to the latter years (2018-2019) where the instrument is operating in 3-point grid mode, with extended latitudinal sun-glint coverage.

While we do not believe that these changes are detrimental to the continued consistency of the timeseries of GOSAT observations, we do feel that it is worth noting as they may have an impact (positively or negatively) on specific applications that a user may wish to use the data for, hence the reason for highlighting them here.

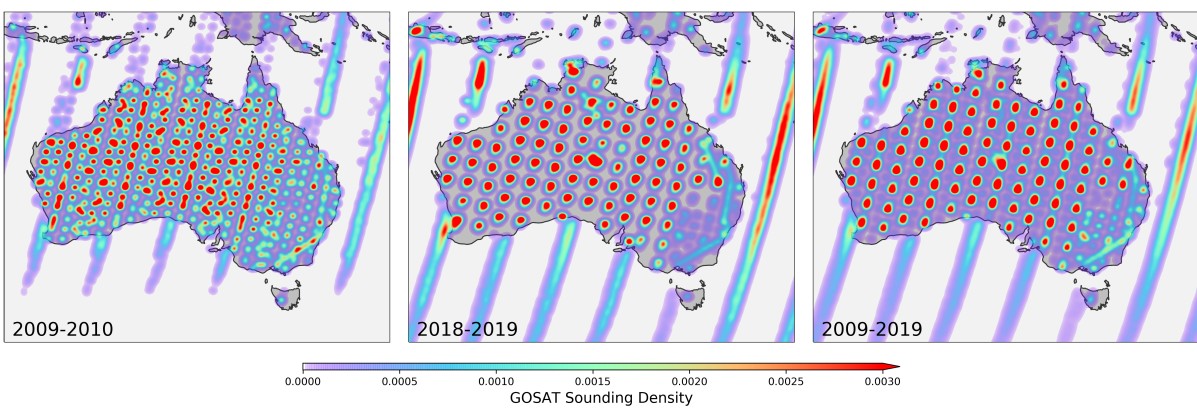

**Figure E2.** Figure contrasting the sampling density pattern over Australia for early in the mission versus recent years. The 5-point sampling grid as used in the early years of the missions was updated to a more stable 3-point grid which continues to be used. The change in pointing mechanism has allowed the ocean sun-glint measurement range to be extended.

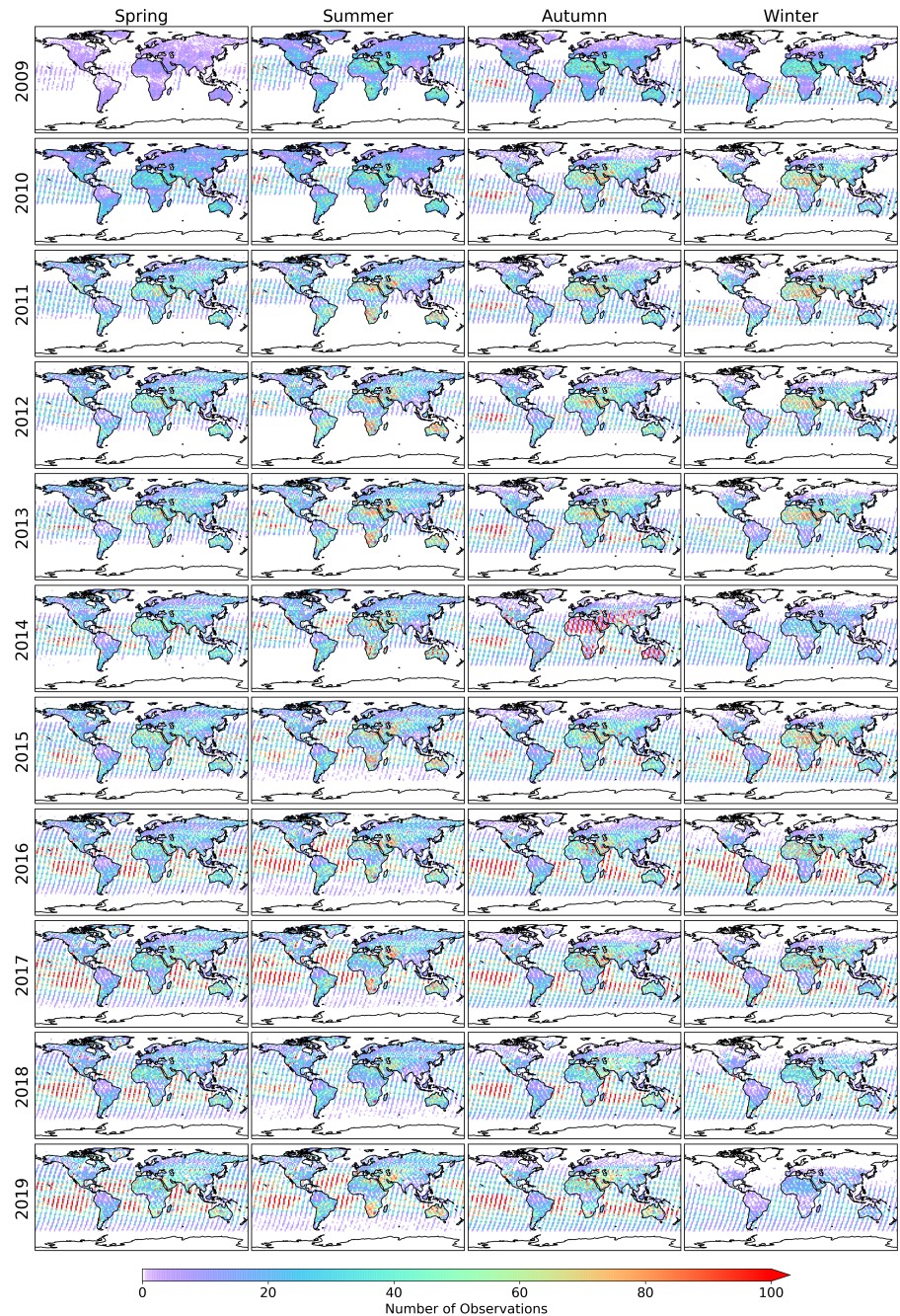

**Figure E3.** Figure showing the number of successful GOSAT XCH$_4$ measurements per 2° latitude-longitude bin for each year/season.

## Appendix F:  TCCON Data

The section provides details and references for the TCCON data used in this study. Table lists all of the TCCON sites used in the study, along with their latitude, when the data record begins and the citation to each specific dataset.

| Site | Latitude | Established | Data Citation |
|------|----------|-------------|---------------|
| Eureka | 80.05°N | July 2010 | Strong et al. (2019) |
| Sodankylä | 67.37°N | January 2009 | Kivi et al. (2014) |
| East Trout Lake | 54.35°N | October 2016 | Wunch et al. (2018) |
| Bialystok | 53.23°N | March 2009 | Deutscher et al. (2014) |
| Bremen | 53.10°N | July 2004 | Notholt et al. (2019) |
| Karlsruhe | 49.10°N | September 2009 | Hase et al. (2015) |
| Paris | 48.85°N | September 2014 | Té et al. (2014) |
| Orleans | 47.97°N | August 2009 | Warneke et al. (2014) |
| Garmisch | 47.476°N | July 2007 | Sussmann and Rettinger (2018) |
| Park Falls | 45.95°N | May 2004 | Wennberg et al. (2017) |
| Rikubetsu | 43.46°N | November 2013 | Morino et al. (2018c) |
| Lamont | 36.60°N | July 2008 | Wennberg et al. (2016) |
| Anmyeondo | 36.5°N | August 2014 | Goo et al. (2014) |
| Tsukuba (125HR) | 36.05°N | December 2008 | Morino et al. (2018a) |
| Saga | 33.24°N | June 2011 | Shiomi et al. (2014) |
| Burgos | 18.53°N | March 2017 | Morino et al. (2018b) |
| Ascension Island | 7.92°S | May 2012 | Feist et al. (2014) |
| Darwin | 12.42°S | August 2005 | Griffith et al. (2014a) |
| Reunion Island | 20.90°S | September 2011 | De Maziere et al. (2017) |
| Wollongong | 34.41°S | May 2008 | Griffith et al. (2014b) |
| Lauder (125 HR) | 45.04°S | February 2010 | Sherlock et al. (2014b) |
| Lauder (120 HR) | 45.04°S | June 2004 | Sherlock et al. (2014a) |

**Table A1.** The TCCON sites used in this study, along with their latitude, when they were established and the citation to the data used.

*Author contributions.* RJP developed and produced the Proxy XCH$_4$ data, performed the analysis and wrote the manuscript. AW assisted in the production of the data. AW, PS, ADN, HB, JA, RBG and NK all contributed to development and analysis at different stages of the processing chain. All authors contributed towards discussion and interpretation of the analysis. PB, FC, PIP and LF provided model data and contributed to the interpretation of the comparisons. All TCCON co-authors provided TCCON data and contributed towards interpretation of the GOSAT-TCCON comparisons.

*Competing interests.* The authors are not aware of any competing interests.

*Acknowledgements.* RJP, HB, AW and LF are funded via the UK National Centre for Earth Observation (NE/R016518/1 and NE/N018079/1). RBG and NK were funded by a Leicester Institute for Space and EO (LISEO) and ESA-Dragon Programme studentship respectively. JSA was funded by an ESA Living Planet Fellowship. We acknowledge funding from the ESA GHG-CCI and Copernicus C3S projects. We thank the Japanese Aerospace Exploration Agency, National Institute for Environmental Studies, and the Ministry of Environment for the GOSAT data and their continuous support as part of the Joint Research Agreement. This research used the ALICE High Performance Computing Facility at the University of Leicester for the GOSAT retrievals and analysis. The TM5-4DVAR CH$_4$ inversions have been supported by ECMWF providing computing resources under the special project "Improve European and global CH$_4$ and N$_2$O flux inversions (2018-2020)".

**TCCON Site Acknowledgements:**

TCCON gratefully acknowledge financial support by ESA within the S5P validation programme. Stations at Park Falls, Lamont and Darwin are supported by NASA. Stations at Tsukuba, Rikubetsu and Burgos are supported in part by the GOSAT series project. Burgos is supported in part by the Energy Development Corp. Philippines. Ascension Island and Garmisch stations have been supported by the European Space Agency (ESA) under grant 4000120088/17/I-EF and by the German Bundesministerium fürr Wirtschaft und Energie (BMWi) under grants 50EE1711C, 50EE1711E and 50EE1711D. We thank the ESA Ariane Tracking Station at North East Bay, Ascension Island, for hosting and local support. The ETL station is funded by CFI/ORF, NSERC, ECCC, and the CSA. The Paris station has received funding from Sorbonne Université, the French research center CNRS, the French space agency CNES, and Région Île-de-France. The Eureka measurements were made at the Polar Environment Atmospheric Research Laboratory (PEARL) by the Canadian Network for the Detection of Atmospheric Change (CANDAC), primarily supported by the Natural Sciences and Engineering Research Council of Canada, Environment and Climate Change Canada, and the Canadian Space Agency. The Anmyeondo station has received funding from the Korea Meteorological Administration Research and Development Program "Development and Assessment of IPCC AR6 Climate Change Scenario" under grant (1365003000). The Réunion Island station is operated by the Royal Belgian Institute for Space Aeronomy with financial support in 2014, 2015, 2016, 2017, 2018 and 2019 under the EU project ICOS-Inwire and the ministerial decree for ICOS (FR/35/IC4) and local activities supported by LACy/UMR8105 – Université de La Réunion. TCCON measurements in Australia are supported by NASA grants NAG5-12247, NNH05-GD07G, Australian Research Council grants LE0668470, DP089468, DP110103118, DP140101552, DP160101598 and FT180100327, as well as the GOSAT series project.

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
