# Peer review of "A Decade of GOSAT Proxy Satellite CH4 Observations"

_Earth System Science Data, 2020_

## Referee Comment (RC1) · Anonymous Referee #1 · 20 Jul 2020

This paper presents the latest GOSAT methane product from U. Leicester including excellent information on how it was produced, data density, validation, and main features. GOSAT methane is a very important Earth Science data product for understanding methane sources and trends, and the previous versions of the Leicester retrieval have been used extensively in the literature. This new product will be eagerly used by the research community. The paper is very well written and a pleasure to read. It could be published as is. I just have a few suggestions for the authors to consider:

1. Abstract: I think it is important to mention that TM5 assimilates NOAA background data, because it is in fact unclear how much of the comparison to GOSAT reflects consistency of GOSAT with NOAA vs. with the TM5 methane budget and transport. This lack of clarity should be acknowledged, not necessarily in the abstract but surely

in the text (it's kind of there, could be made stronger).

2. Page 4, around line 5: mention that GOSAT also operates in target mode. Hasn't it been doing so increasingly in recent years? This provides additional info but also disrupts the time series. I also don't think that the time series have been kept at consistent locations for the 10 years – they seem to shift. But the authors would know for sure.

3. Page 5, line 26: Maasakkers et al. 2019 comment on OH variability but attribute most of the 2010-2015 GOSAT trend to tropical wetlands and livestock.

4. Page 6, Section 4: I'm surprised that the role of aerosols (e.g., dust plumes) is not mentioned anywhere in this section.

5. Page 9, Section 7: see comment in the abstract. In comparisons over broad regions like in Figure 12 one wonders how much of the fit is due to the assimilation of NOAA data in the model.

6. Page 21, line 12: two good references for model biases in the stratosphere are Patra et al. ACP 2011 and Saad et al. ACP 2016.

7. Page 21, line 14: Turner et al. ACP 2015 also have a quadratic polynomial correction for model-GOSAT bias vs latitude.
* * *

---

## Referee Comment (RC2) · Anonymous Referee #2 · 21 Sep 2020

This is a descriptive and helpful paper on GOSAT XCH4 proxy product developed by University of Leicester.

Two main comments: First, why have there been 9 Versions of the GOSAT XCH4 proxy product? A table documenting the version, release date, and changes, would be very helpful. In addition, there is no forward looking discussion..., what can we expect for version 10 and upward? What remains or is needed in future, e.g., how will constellations be integrated, or GOSAT2. What is mission lifetime for GOSAT given degradation of instruments, orbit, etc.

For the presented v9, a table summarizing key components is needed, such as spatial and temporal parameters, overpass time, accuracy, etc. Currently, the reader has to dig around for this key information.

[Figure]

Introduction - Mention/acknowledge there is also GOSAT-2 - Mention the nickname, IBUKI

Section 2.0: - What were the original measurement requirements – please list these as they provide a context for how well this product performs. - Where is the Project Science Office located – mention this as its important to acknowledge the primary data management group.

Section 3.2 - Refer to Sentinel-5 nomenclature when TROPOMI is mentioned - Also, PRISMA and HISUI should be mentioned for their CH4 retrieval potential.

Typo – systemtically

Fig 10 – given how small these figures are, and the size of the grid cel used to represent the XCH4 concentration, the images give the reader a false sense of coverage. I would recommend a new, similar paneled, figure that has something like # of observations, or the cloud mask, to highlight the geographic coverage issue more clearly.

---

## Referee Comment (RC3) · Anonymous Referee #3 · 4 Oct 2020

<General Comments> Annual growth-rate of atmospheric methane-concentration is not constant. There are various sources, of which emission amount have large uncertainties. Long term and global data of proxy CH4 concentration measured with a single space-borne instrument are valuable. The dataset is well validated with TCCON and models and its application such as flux estimation is well described. I recommend publication after minor modification.

<Specific Comments> (1) Page 3, Line 15, "This version (ver9)" Are the full-physics and proxy algorithms updated simultaneously? Brief explanation of version up history and its improvements will help readers' understanding.

(2) Page 4, Line 12, "Switch to the secondary pointing mechanism" GOSAT sampling pattern has changed since January 2015 when the unstable primary pointing mech-

anism has been changed. The secondary mechanism has much better performance of settling down. Both pointing fluctuation and bias had been removed. It also has wider pointing range in the along track direction. Therefore, more target observations for large emission sources have been allocated instead of grid observation with the nominal 3-point cross-track scan mode. Over the ocean, latitudinal range of glint observation has been widened. For 11-year GOSAT operation, this is the largest event that had affected the performance. Addition of description will be helpful to understand the description on Figure 4 in page 10.

(3) Page 8, Line 21, "sounding specific a priori information" and Page 9, Line 15, "Spectral dispersion" Do authors retrieve wavelength every time by fitting GOSAT spectral to the simulation spectra?

<Technical Corrections> (1) pages 28-32, references A few discussion papers of AMTD and ACPD are referred. They seem to be reviewed and published as AMT and ACP papers.

---

## Author Comment (AC1) · 21 Oct 2020

**Response to Review Comments for *A Decade of GOSAT Proxy Satellite CH₄ Observations**

Parker et al.

October 21, 2020

**Firstly, we would like to express our gratitude to the editor and reviewers for providing a thorough review of our paper. We appreciate their efforts, especially in these difficult times.**

Please see the supplement for additional figures referred to below.

**1  Response to RC1 Review Comments**

(Original Comment, Our Response, New Manuscript Text)

1. Abstract: I think it is important to mention that TM5 assimilates NOAA background data, because it is in fact unclear how much of the comparison to GOSAT reflects consistency of GOSAT with NOAA vs. with the TM5 methane budget and transport. This lack of clarity should be acknowledged, not necessarily in the abstract but surely in the text (it's kind of there, could be made stronger).

To explain, our comparisons to model data are to ensure that we are in broad agree-

ment with simulations at the level where further scientific investigation (e.g. flux inversions) would make sense to conduct. Comparisons to a free-running unconstrained $CH_4$ simulation would likely show offsets/differences and it would then be unclear if these were due to the satellite data or the model simulation. By having good overall agreement to both TCCON and the NOAA-constrained model simulations, we believe this indicates the utility of our data for use in such studies. We will add additional clarification to the manuscript text in Section 7.

We have chosen to compare against model simulations that are both widely used within the community and that have already assimilated NOAA surface measurements. The reasoning for this is that any overall differences as might be seen from free-running model simulation are removed and we can clearly state the consistency of our dataset with the NOAA network. By proving good overall agreement to both TCCON measurements (Section 5) and the NOAA-constrained model simulations we believe this indicates the utility of our data for use in further scientific analysis.

2. Page 4, around line 5: mention that GOSAT also operates in target mode. Hasn't it been doing so increasingly in recent years? This provides additional info but also disrupts the time series. I also don't think that the time series have been kept at consistent locations for the 10 years – they seem to shift. But the authors would know for sure.

We have included an additional figure which we believe is informative and addresses this point. The GOSAT measurement strategy/capability has indeed changed over time, however we do not believe that the timeseries has been "disrupted" in any significant way. As discussed in RC3, the switch to the secondary pointing mechanism has had an effect on the coverage, extending the ocean glint range and the number of target observations. We discuss this in the manuscript and highlight it with Figure 10. This additional figure (*Response Supplement Figure 1*) provides further details, particularly relating to the 3-point repeat grid vs target observations. We have added this to a new appendix section detailing the sampling/measurement details and also included a second figure (*Response Supplement Figure 2*) that we believe is informative.

This section provides details on how the GOSAT measurement strategy has evolved over time. GOSAT initially operated primarily on a regular 5-point grid, revisiting the same grid point. This changed over time to a 3-point grid in order to reduce pointing uncertainty at the extreme angles when using the primary pointing mechanism. However, one consequence of a regular grid is that this resulted in a limited number of observations over some regions, especially islands, where the regular grid point would fall over the ocean. Thanks to the switch to the secondary pointing mechanism, several changes to the sampling strategy were possible.

Firstly, GOSAT is now capable of targeting more specifically and "follows" coastlines in a more efficient manner. The example in Figure 1 shows the change in sampling location over Indonesia, contrasting 2011 to 2017. Although the exact same grid location is not revisited in the same way, overall there are both more successful measurements and a better geographic coverage.

Secondly, GOSAT is now capable of a wider pointing range and subsequently, the latitudinal range of ocean sun-glint observations have been extended (as observed in Figure 10). Figure 2 provides example comparisons of the measurement density over Australia between the start of the mission (2009-2010) when the instrument was primarily operating in 5-point grid mode to the latter years (2018-2019) where the instrument is operating in 3-point grid mode, with extended latitudinal sun-glint coverage.

While we do not believe that these changes are detrimental to the continued consistency of the timeseries of GOSAT observations, we do feel that it is worth noting as they may have an impact (positively or negatively) on specific applications that a user may wish to use the data for, hence the reason for highlighting them here.

3. Page 5, line 26: Maasakkers et al. 2019 comment on OH variability but attribute most of the 2010-2015 GOSAT trend to tropical wetlands and livestock.

We've adjusted this reference in the manuscript to cover both aspects.

[Figure]

4. Page 6, Section 4: I'm surprised that the role of aerosols (e.g., dust plumes) is not mentioned anywhere in this section.

We have added in explicit mention of aerosol and why the proxy retrieval method mitigates a lot of the aerosol considerations that would be required for the "full physics" algorithm.

This means that moderate aerosol scattering does not adversely impact upon the retrieval, resulting in a higher number of high-quality $XCH_4$ observations compared to the full physics approaches where much stricter post-filtering are often required. This is especially useful over the tropics, where moderate aerosol or cirrus effects can limit the coverage of full-physics methods but affect the proxy approach far less severely.

5. Page 9, Section 7: see comment in the abstract. In comparisons over broad regions like in Figure 12 one wonders how much of the fit is due to the assimilation of NOAA data in the model.

See response above for abstract comment.

6. Page 21, line 12: two good references for model biases in the stratosphere are Patra et al. ACP 2011 and Saad et al. ACP 2016.

We will include these in the manuscript.

7. Page 21, line 14: Turner et al. ACP 2015 also have a quadratic polynomial correction for model-GOSAT bias vs latitude.

Noted. We will include reference to this in the appropriate section.

**2  Response to RC2 Review Comments**

(Original Comment, Our Response, New Manuscript Text)
Two main comments: First, why have there been 9 Versions of the GOSAT XCH$_4$ proxy product? A table documenting the version, release date, and changes, would be very helpful.

We have now included a detailed breakdown of the previous versions of the UoL GOSAT Proxy XCH$_4$. We initially omitted this as we wanted to focus on v9.0 but we accept that this is a useful table and puts the current data in context.

Table 1 (*Response Supplement Figure 3*), provides details of the evolution of the University of Leicester GOSAT Proxy XCH$_4$ data product. Entries include the version number, the project that the data was generated for, the version of the GOSAT L1B data used, the time period covered by the data, whether ocean sun-glint data was generated, comments relating to changes/updates from previous versions and peer-reviewed publications that we are aware of that used the data. For the ESA GHG-CCI project, we also indicate which versions were officially delivered as part of the Climate Research Data Packages through the project. All Copernicus C3S versions were delivered to the Copernicus Climate Data Store.

In addition, there is no forward looking discussion..., what can we expect for version 10 and upward? What remains or is needed in future, e.g., how will constellations be integrated, or GOSAT2. What is mission lifetime for GOSAT given degradation of instruments, orbit, etc.

We have also included a statement regarding the future evolution of the product. However, we do not feel it is our place to comment directly on the future of the GOSAT mission itself as that is the responsibility of JAXA and we would not want our response to be minconstrued as any sort of official position.

Despite GOSAT-1 having a planned mission lifetime of 5 years, it continues to successfully perform measurements 11 years after launch. GOSAT-2 was launched in October 2018 (Suto et al, 2020) and will continue the legacy of the GOSAT-1 mission. GOSAT-2 offers several opportunities for development related to the dataset we describe here.

Primarily, it ensures that should GOSAT-1 cease operation, the valuable decade-long timeseries of observations can continue to be extended via GOSAT-2. With a significant overlap in time between the two missions, consistency between the two missions can be assured, albeit with significant future work/development.

In addition, GOSAT-2 has additional capabilities, namely the possibility of measuring carbon monoxide (CO). By measuring $CO_2$, $CH_4$ and CO simultaneously from the same instrument, GOSAT-2 would allow the extension of studies examining biomass burning combustion, leading to constraints on fire emission ratios as have been performed previously for GOSAT-1 (Ross et al., 2013; Parker et al., 2016).

For the presented v9, a table summarizing key components is needed, such as spatial and temporal parameters, overpass time, accuracy, etc. Currently, the reader has to dig around for this key information

Table summarising product characteristics has been included.

Table 1 summarises the key characteristics of the University of Leicester GOSAT Proxy $XCH_4$ data, including the spatial and temporal extent that the dataset covers, the total number of measurements and their evaluation against TCCON.

| Attribute | Value |
|---|---|
| Temporal Extent | 2009-2019 |
| Spatial Extent | Global (56.3°S - 83.5°N) |
| Total Number of Measurements | 4.6 million |
| Footprint Size | 10.5km (at nadir) |
| Overpass Time (at Equator) | ~13:00 Local Solar Time |
| Bias (vs TCCON) | 0 ppb (after global bias correction of 9.06 ppb) |
| Precision (vs TCCON) | 13.72 ppb |

**Table 1.** Table summarising the key characteristics of the University of Leicester GOSAT Proxy $XCH_4$ data.

Introduction - Mention/acknowledge there is also GOSAT-2 - Mention the nickname, IBUKI

We have included both of these points either here or above.

GOSAT is nicknamed *"Ibuki"*, meaning *breath* in Japanese, highlighting that its mission involves monitoring the breathing of the planet, through measurement of the carbon cycle.

Section 2.0: - What were the original measurement requirements – please list these as they provide a context for how well this product performs.

We have quoted the original measurement requirements as outlined in Kuze et al., 2009. Furthermore, we also refer to the User Requirements developed during the ESA CCI project.

Kuze et al. (2009) state a target relative accuracy of 2% for $CH_4$ over 3 month averages at 100-1000 km spatial scales. The ESA GHG-CCI User Requirements Document (URD) specify goal (G), breakthrough (B) and threshold (T) requirements, with the goal requirements being the most stringent (Buchwitz et al, 2017). For $XCH_4$ precision, these values are 9, 17 and 34 ppb respectively. We discuss in Section 5 how we exceed these breakthrough requirements.

The single measurement precision of 13.72 ppb comfortably exceeds the precision breakthrough requirement of 17 ppb (Buchwitz et al, 2017) indicating that it "would result in a significant improvement for the targeted application"

Where is the Project Science Office located – mention this as its important to acknowledge the primary data management group.

We are extremely grateful for the help and support we have received from the entire GOSAT project. We acknowledge this in the acknowledgements section. We have also reached out to the GOSAT Project Office directly to ensure that they are happy with the current statement and they have confirmed that they are.

Section 3.2 - Refer to Sentinel-5 nomenclature when TROPOMI is mentioned - Also, PRISMA and HISUI should be mentioned for their CH$_4$ retrieval potential.

...continue to be performed from new missions such as TROPOMI onboard Sentinel 5-Precursor and...

We have included menton of PRISMA and HISUI in the forward looking discussion.

A strong focus of future CH$_4$-measuring satellites will be to examine anthropogenic emission sources at very high spatial resolution (e.g. PRISMA (Stefano et al, 2013);HISUI (Matsunaga et al., 2018);ENMAP (Guanter et al., 2015)), particularly relating to monitoring of the oil and gas industry. However, many scientific challenges and questions remain regarding the long-term CH$_4$ behaviour and the response to a changing climate. For this reason, a long-term, consistent climate-ready data record as we present here is of continued importance.

Typo – systemtically

Fixed in manuscript.

Fig 10 – given how small these figures are, and the size of the grid cell used to represent the XCH$_4$ concentration, the images give the reader a false sense of coverage. I would recommend a new, similar paneled, figure that has something like # of observations, or the cloud mask, to highlight the geographic coverage issue more clearly.

Additional figure *Response Supplement Figure 4* showing number of observations. This has been incorporated into the new appendix section detailing the sampling/measurement details.

Figure 4 shows the number of successful GOSAT XCH$_4$ measurements per 2° latitude-longitude bin for each year/season.

**3  Response to RC3 Review Comments**

(Original Comment, Our Response, New Manuscript Text)

**Specific Comments**

(1) Page 3, Line 15, "This version (ver9)" Are the full-physics and proxy algorithms updated simultaneously?

This paper deals purely with the proxy retrieval algorithm and does not relate to the full-physics algorithm at all. These algorithms are developed and processed separately, with the $CH_4$ full-physics being closely tied to the $CO_2$ full-physics.

Brief explanation of version up history and its improvements will help readers' understanding.

Please see response to RC2. We have included a detailed version history in the appendix.

(2) Page 4, Line 12, "Switch to the secondary pointing mechanism" GOSAT sampling pattern has changed since January 2015 when the unstable primary pointing mechanism has been changed. The secondary mechanism has much better performance of settling down. Both pointing fluctuation and bias had been removed. It also has wider pointing range in the along track direction. Therefore, more target observations for large emission sources have been allocated instead of grid observation with the nominal 3-point cross-track scan mode. Over the ocean, latitudinal range of glint observation has been widened. For 11-year GOSAT operation, this is the largest event that had affected the performance. Addition of description will be helpful to understand the description on Figure 4 in page 10.

We have added some of this very useful description to the manuscript text. We have also included a new appendix section and additional figures in response to the other review comments that directly relate to this topic. Please see response to RC1.

(3) Page 8, Line 21, "sounding specific a priori information" and Page 9, Line 15, "Spectral dispersion" Do authors retrieve wavelength every time by fitting GOSAT spectral to the simulation spectra?

Yes. We do state in the text that our retrieval state-vector contains components related to the spectral dispersion but we will make this more specific.

...allowing us to explicitly fit the wavelength for each spectra independently.

3.1 Technical Corrections

(1) pages 28-32, references A few discussion papers of AMTD and ACPD are referred. They seem to be reviewed and published as AMT and ACP papers.

We have updated all citations in the manuscript where a final published version of the reference has now been published.

Please also note the supplement to this comment:
https://essd.copernicus.org/preprints/essd-2020-114/essd-2020-114-AC1-supplement.pdf

**Supplement:**

**Response to Review Comments for *A Decade of GOSAT Proxy Satellite CH$_4$* Observations**

Parker et al.

October 14, 2020

**This supplement contains the additional figures that go along with our Author Response.**

[revised manuscript text omitted]